# Kite as a Sensor: Wind and State Estimation in Tethered Flying Systems

Oriol Cayon[1], Simon Watson[1], and Roland Schmehl[1]

[1]Faculty of Aerospace Engineering, Delft University of Technology, 2629 HS Delft, the Netherlands

**Correspondence:** Oriol Cayon (o.cayon@tudelft.nl)

**Abstract.** Airborne wind energy systems (AWESs) leverage the generally less variable and higher wind speeds at increased altitudes by utilizing kites, with significantly reduced material costs compared to conventional wind turbines. Energy is commonly harnessed by flying crosswind trajectories, which allow the kite to achieve speeds significantly higher than the ambient wind speed. However, the airborne nature of these systems demands active control and makes them highly sensitive to changes in wind conditions, making accurate wind measurements essential for steering the kite along its optimal trajectory. This paper presents an advanced sensor fusion technique based on an iterated extended Kalman filter (EKF) for state and wind estimation for AWESs. By integrating position, velocity, tether force, and reeling speed, this method provides accurate estimations of system dynamics, including kite orientation and tether shape. The estimates of the wind speed and direction are compared to lidar measurements, showing good agreement across various atmospheric conditions, with 10-minute averaged root mean squared error (RMSE) values below $1 \ \mathrm{ms}^{-1}$ and $5°$, respectively. The results demonstrate that this approach can effectively capture the transient dynamics of atmospheric wind using sensors typically already present in AWESs, making it suitable for supervisory control strategies and ultimately enhancing energy efficiency and system reliability across diverse atmospheric conditions.

## 1 Introduction

Airborne wind energy systems (AWESs) harness wind energy with tethered aerial devices, substantially reducing material usage compared to conventional wind turbines by employing one or more tethers and a flying apparatus instead of towers and blades. This reduction has the potential to lower the costs associated with wind energy production but also allows for the exploitation of higher-altitude winds, which are generally less variable and of a higher average speed than those at ground level.

Despite these advantages, AWESs face significant challenges, particularly regarding system robustness against the complexities of atmospheric wind dynamics. The motion of the kite during crosswind flight is strongly influenced by the wind, which largely dictates the flight speed and the tether force. This dependency makes the system highly susceptible to changes in wind speed and direction. For soft kites, the low mass of the tensile structure allows for rapid adaptation to changes in wind speed, making this type of kite particularly sensitive to turbulence and gusts. Therefore, a detailed understanding of wind dynamics at the operational altitudes is crucial.

Above the well-studied surface layer, logarithmic wind profiles may not accurately represent the variation of wind speed with height, and phenomena such as wind veer and low-level jets become increasingly significant (Kalverla et al., 2017). As a result, it is crucial to adapt the kite's trajectory to these specific higher-altitude wind conditions, which necessitates reliable wind measurements. This need for accurate wind data can be met through remote sensing devices such as lidar or sodar (Sommerfeld et al., 2019; Khan and Tariq, 2018; Watson, 2023b), with lidar being more frequently used except in cold, clear-air climates

where its performance may be limited, or by using the kite itself as a sensor.

Using the aerial device as a sensor eliminates the need for additional equipment beyond what is already used for kite control and allows wind velocity information to be integrated into a supervisory control strategy for the AWES. This integration enables the system to adjust the flight trajectory in response to changing wind conditions, optimizing performance and aiding in high-level decisions, whether to take-off or to land.

A common method for determining ambient wind conditions at the kite involves mounting flow sensors on the kite to measure the apparent wind speed and direction. Pitot tubes are frequently used, in varying configurations. Five-hole probes and single-hole probes combined with two flow vanes can measure the three-dimensional velocity vector of the apparent wind (Elfert et al., 2024; Borobia-Moreno et al., 2021; Oehler and Schmehl, 2019), allowing for the determination of ambient wind speed, provided the kite's velocity is known. Simpler setups with single-hole probes alone (Vlugt et al., 2013; Borobia et al.,

2018) or with one vane (Schelbergen, 2024) are also used but do not capture the full three-dimensional velocity, limiting their capability.

However, this approach comes with significant challenges. The accuracy of the measurements depends heavily on the sensor's position, mounting method, and regular maintenance and recalibration. For soft kites, which undergo substantial deformation during flight along with vibrating bridle lines and fluttering membrane, the collected data can become excessively noisy

and unreliable (Dunker, 2018; Leuthold, 2015). These challenges, particularly for tensile, lightweight kite systems, highlight the need for more advanced approaches.

One effective solution is sensor fusion, which combines data from multiple sensors integrated into the AWES to provide a more robust and consistent representation of the kite state and wind characteristics. Sensor fusion relies on a time-dependent model that represents the system's dynamics. By integrating sensor data within this model, the method enhances the reliability

of the measured states and can also serve to estimate quantities such as wind speed and direction. Depending on the model's complexity, the estimated state may encompass not only the kite's position and velocity but also its aerodynamic characteristics and the tether sag, resulting in a more comprehensive understanding of the system's dynamics. For clarity, the notation and symbols used throughout this paper are summarized in Appendix A.

Numerous studies have investigated sensor fusion techniques for AWE state and wind estimation (see Table 1). For instance,

Fagiano et al. (2013, 2014) proposed an extended Kalman filter (EKF) to estimate the kinematics of a tethered aircraft, using a purely kinematic model and evaluating various sensor configurations, including satellite-based global positioning system (GPS) and tether angle measurements. Similarly, Polzin et al. (2017); Wood et al. (2018) explored configurations incorporating inertial measurement units (IMUs), tether angles, and camera tracking, introducing a novel kinematic model that accounts for tether dynamics and sag through time delays and ground velocity differences. This study also addressed kite steering delays by

**Table 1.** Summary of studies on sensor fusion for AWESs.

| Study | Filter Type | Minimum Required Measurements | Kite Model | Tether Model |
|---|---|---|---|---|
| Fagiano et al. (2013, 2014) | EKF | Position, acceleration, orientation, orientation rates | Kinematic | Straight & Inelastic |
| Polzin et al. (2017); Wood et al. (2018) | EKF | Position, orientation, orientation rates | Kinematic | Straight & Inelastic |
| Freter et al. (2020) | Adaptive EKF | Position, acceleration, tether force | Kinematic | Straight & Inelastic |
| Williams et al. (2008) | Square-root UKF | Position, tether force and direction, tether length and reeling speed | Point Mass | Straight & Inelastic |
| Ranneberg (2013) | Square-root UKF | Position, tether force, tether length, steering input | Point Mass | Straight & Inelastic |
| Schmidt et al. (2017, 2020) | EKF | Position, velocity, wind speed and direction (reference height), tether force, tether length and reeling speed, steering input | Point Mass | Straight & Inelastic |
| Borobia et al. (2018); Borobia-Moreno et al. (2021) | EKF | Position, velocity, angular velocities, magnetic field, airspeed magnitude, tension at the bridles, kite center of mass | Rigid Body | Straight & Inelastic |
| Current Study | Iterated EKF | Position, velocity, tether force, tether reeling speed | Point Mass | Curved & Elastic (Quasi-Static) |

modelling the yawing motion with a linear turn rate law. To further address tether sag, Freter et al. (2020) proposed an adaptive Kalman filter with variable weights based on tether force, combining data from load cells, tether angles, IMUs, and camera tracking, which effectively reduced estimation errors under sagged conditions. However, these approaches do not estimate the wind velocity, which requires the inclusion of forces in the modelling to capture the wind impact on system behaviour.

In Williams et al. (2008), an unscented Kalman filter (UKF) was proposed as a state estimator for both the kite's state and
wind conditions within a non-linear tracking control framework. The simulation results demonstrated that the UKF performed well when assuming a straight tether and was robust against noise. However, the wind velocity estimates were found to be the most sensitive to noise.

Ranneberg (2013) used a UKF with a Lagrangian dynamic model that incorporates the forces on the kite, assuming constant aerodynamic coefficients, as well as rotational inertia of the drum. The results were validated against simulation data and
experimental measurements, including pulsed lidar measurements at various heights, demonstrating that measuring the kite's position and the tether force is sufficient for estimating wind speed, although the vertical component of the wind velocity was assumed to be zero. Schmidt et al. (2017, 2020) proposed a similar approach using an EKF with a Lagrangian dynamic

model but without including drum inertia. Their model adds measured wind speed and direction at a reference altitude, which works well for their case study of a kite flying at low altitudes but can introduce errors when the kite flies at higher altitudes. Nevertheless, the model is validated against experimental and simulation data, showing the potential of the approach.

In Borobia et al. (2018); Borobia-Moreno et al. (2021), a more complex model is employed to estimate the full state of the kite, including the ambient wind conditions and aerodynamic forces and moments. However, this increased complexity necessitates a significantly larger number of measurements. These measurements include data from GPS and IMU sensors, Pitot tube, including a five-hole Pitot tube, wind sensors on the ground and several load cells installed in the bridle line system of the kite.

A common limitation of existing methods is the assumption of a straight and inelastic tether, which can introduce errors when the tether force is low and the real tether sags. Additionally, soft kites with a suspended robotic control unit are affected by the inertia of the suspended unit, impacting orientation and dynamics (Roullier, 2020). The current study addresses these limitations by employing a more comprehensive model that accounts for both tether sag and KCU inertial forces, enabling accurate wind velocity estimation without relying on direct airflow measurements at the kite. Specifically, we employ an iterated EKF that models the wing as a point mass and incorporates a quasi-static tether model (Williams, 2017). To account for the inertial effects of the KCU, an additional point mass is included between the tether and the wing, resulting in a two-point representation of the kite (Schelbergen and Schmehl, 2024).

Validation against independent sensor data confirms the accurate reconstruction of the kite state, while comparison with lidar measurements demonstrates the model's capability to estimate ambient wind conditions. During the reel-out phase, the 10-minute averaged wind speed error remained below 1 $\mathrm{ms}^{-1}$, and the wind direction error below 5°. Although the study focuses on soft kites, the methodology is equally applicable to rigid-wing devices.

The results are presented for two kite prototypes, the V3 and the V9, both being leading-edge inflatable kites operated by Kitepower B.V., with flattened wing areas of 25 and 60 m$^2$, respectively. The V3 kite has been extensively studied in prior research and has served as a reference model (Oehler and Schmehl, 2019; Viré et al., 2022; Cayon et al., 2023; Poland and Schmehl, 2023; Schelbergen and Schmehl, 2024). In contrast, the V9 kite, representing Kitepower's current commercial prototype, is primarily used in the present study to showcase the method's ability to predict wind velocities at the kite. This analysis is based on an unprecedented dataset that combines high-quality AWES operational data with high-resolution wind measurements from profiling lidar. The AWES data include measurements of kite position, velocity, orientation, and accelerations; tether tension, length, and reeling speed; as well as airflow measurements from a Pitot tube and wind vanes, while the lidar data provide vertical wind profiles with 20 m spatial resolution and temporal sampling at either 1 Hz or 1-minute intervals, depending on the dataset. Together, these datasets enable a comprehensive analysis of wind-kite interactions across various atmospheric conditions.

The remainder of this paper is organized as follows. Sect. 2 provides an overview of the AWE system, Sect. 3 discusses the experimental sensor setup, Sect. 4 details the filter design and sensor calibration, Sect. 5 presents the results and analysis, and Sect. 6 concludes with implications and future research directions.

## 2 System overview

The main components of ground-generation soft kite AWESs are the ground station, tether, and kite, which consists of the wing, bridle system, and suspended kite control unit (KCU). The kite flies in cyclic patterns to generate electricity, alternating between traction and retraction phases. During the traction phase, the kite performs crosswind manoeuvres, reeling out the tether and driving a drum connected to a generator. Once the tether reaches its maximum length, the system reverses, operating the generator as a motor to reel in the tether. During this phase, the kite is flown at a lower angle of attack, producing less force and allowing for an efficient reel-in.

In Fig. 1, the key components of a soft kite for airborne wind energy harvesting are shown alongside commonly used sensors. The kite is controlled by the KCU, which connects the bridle line system and the tether. The front bridle lines transmit most of the aerodynamic force from the wing, while the rear bridle lines are used to actuate the wing through the KCU and the steering and depower tapes. The steering tape deforms the wing asymmetrically by modifying the length of the steering lines to initiate turns, while the depower tape symmetrically adjusts the steering lines to pitch the wing relative to the tether. This pitch is quantified by the depower angle $\alpha_{\mathrm{d}}$ (see Fig. 1), explained in more detail below. Increasing this depower angle directly influences the kite's aerodynamic performance and it is mainly used to reel in the kite efficiently.

The aerodynamics of the kite are generally characterized by the angle of attack at the centre section of the wing, $\alpha_{\mathrm{w}}$. However, directly measuring this angle is challenging because it would require isolating the inflow at the kite from the flow disturbances caused by its aerodynamics, which can be difficult to achieve reliably in practice. Consequently, sensors are more commonly installed on the bridle lines (see Fig. 3) or on the tether below the KCU.

The bridle angle of attack $\alpha_{\mathrm{b}}$ is defined as the angle relative to the plane perpendicular to the power lines (see Fig. 1) and is related to the angle at the wing by

$$\alpha_{\mathrm{w}} = \alpha_{\mathrm{b}} - \alpha_{\mathrm{d}}, \tag{1}$$

where the depower angle, $\alpha_{\mathrm{d}}$, is approximated as a linear function of the depower input, $u_{\mathrm{p}}$. This relationship includes an initial offset, $\alpha_{\mathrm{d},0}$, corresponding to the powered kite state,

$$\alpha_{\mathrm{d}} = \alpha_{\mathrm{d},0} + \Delta\alpha_{\mathrm{d}} \, u_{\mathrm{p}}, \tag{2}$$

where $u_{\mathrm{p}}$ ranges from 0 to 1, and $\Delta\alpha_{\mathrm{d}}$ represents the change in angle of attack from the fully powered to the depowered state. The depower angle $\alpha_{\mathrm{d}}$ is typically determined experimentally by manually adjusting the depower tape length to achieve a balance between performance and controllability. In previous works (Oehler and Schmehl, 2019; Schelbergen, 2024), the variation of this angle was estimated using a simple geometric model that relates the length of the depower tape to the depower angle. However, with the integration of the tether model into the EKF, this angle can now be estimated directly from the measurements, as detailed in Sect. 5. Finally, the bridle angle of attack can be translated to the tether angle of attack, $\alpha_{\mathrm{t}}$, defined as the angle to the perpendicular of the tether direction, by

$$\alpha_{\mathrm{t}} = \alpha_{\mathrm{b}} + \lambda_0 + \theta_{\mathrm{k}}, \tag{3}$$

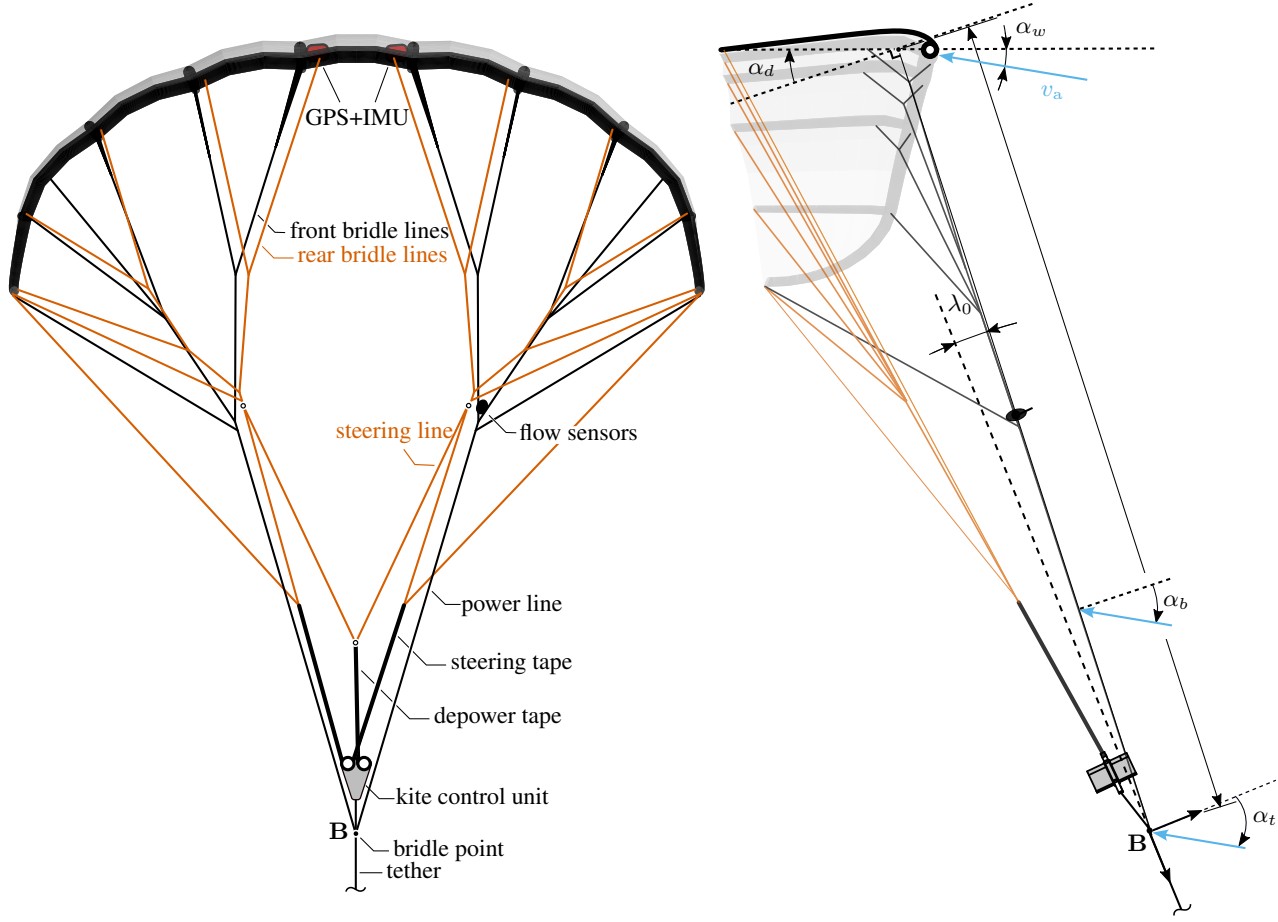

**Figure 1.** Illustration of system components and sensor setup. Adapted from Oehler and Schmehl (2019). The V3 kite geometry shows the bridle point **B** below the KCU, unlike the V9 geometry where the KCU is at the bridle point.

where $\lambda_0$, defined as the angle between the tether and the power lines, depends on the aerodynamic load distribution between the front and rear bridle lines, and $\theta_\mathrm{k}$ is the kite pitch with respect to the tether caused by the KCU weight and inertia. $\lambda_0$ has been found to be relatively constant for a set depower setting but is highly dependent on changes in the depower tape length (Oehler and Schmehl, 2019).

## 3 Sensor setup

Measuring the state of AWESs presents significant challenges, particularly for soft kites, which, by their nature as tensile membrane structures experience substantial deformations during flight, along with vibrating components and high accelerations during turning manoeuvres. Understanding the accuracy and limitations of each sensor within its specific installation context is critical for developing an effective fusion technique. This understanding enables the design of a sensor fusion model relying on the most trustworthy sensors.

The following is a breakdown of the key sensors used in the analysed datasets, along with a qualitative discussion of their advantages, limitations, and considerations for accurate measurements:

- Load Cells: The force exerted by the tether is measured using a load cell installed on one of the pulleys guiding the tether at the exit point of the drum (see Fig. 2). The measured force is projected onto the tether direction using the known pulley geometry and the measured elevation angle of the outgoing tether. This setup can yield accurate results, provided it is correctly calibrated (Hummel et al., 2018). One could also consider installing load cells in the kite's bridle lines to measure the force distribution (Oehler et al., 2018; Borobia-Moreno et al., 2021). However, this approach significantly increases setup times and also the risks of failures and large inaccuracies in the measurements.

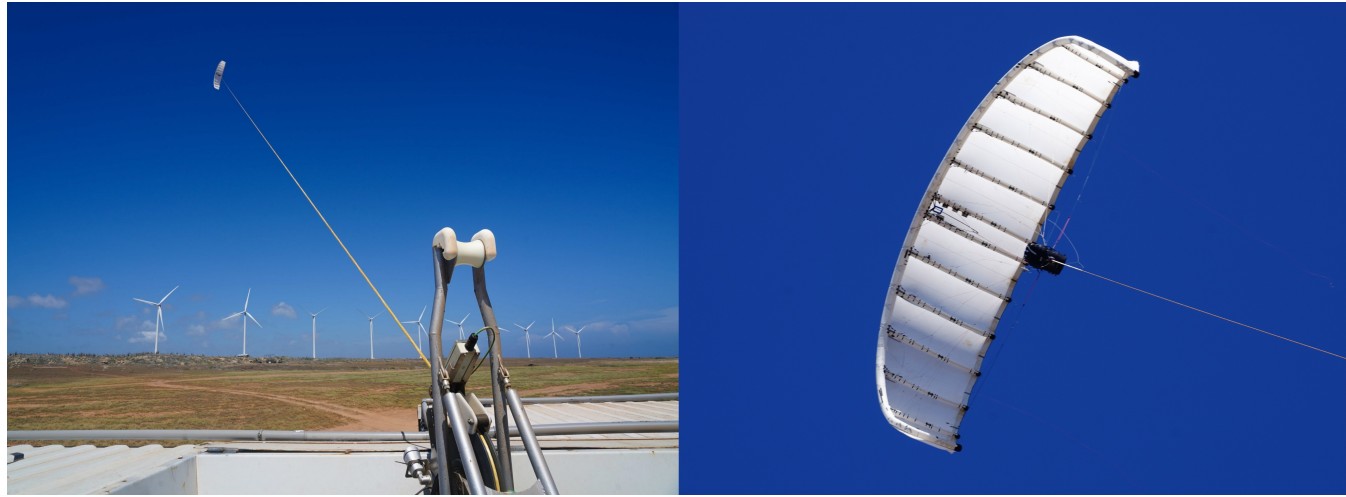

**Figure 2.** The V9 kite in flight: view from the ground station (left), and close-up showing the suspended control unit (right). Photos courtesy of Kitepower B.V.

- Tether length: The length of the tether is measured through the reeling mechanism of the kite, where an incremental angular encoder on the drum determines the rotational speed. From this, the deployed tether length is calculated (Vlugt et al., 2013). This method allows for precise measurements of the reeling speed and the tether length, particularly if the initial offset of the tether length is correctly identified and accounted for, or in the case of the current model, if the tether length offset is also included in the filter.

- Tether angles: The elevation and azimuth angles of the tether can be measured at the ground station using magnetic angular encoders. When combined with the tether length, these measurements can be used to estimate the kite position (Peschel, 2013). The assumption of a straight tether generally holds during reel-out operations when the tether is under high tension, resulting in relatively accurate position estimates (Fagiano et al., 2013). However, the tether generally sags during reel-in due to reduced tension, leading to potential inaccuracies unless the sag is taken into account.

- GPS: These sensors typically provide an accuracy of 1–2 meters, which can be improved to centimeter-level accuracy with the use of real-time kinetic (RTK) positioning (PX4, 2025). Moreover, these measurements are not significantly affected by the deformation of the soft kite, as the displacement at the sensor mounting point is only a few centimeters. This makes GPS a reliable source of position and velocity information. However, GPS signals can suffer from issues such as signal loss, particularly during high-acceleration maneuvers (Vlugt et al., 2013).

- IMU: The placement of an IMU sensor on one of the inflatable tubes (see Fig. 3) of the kite has several implications. The sensor data is affected by the in-flight deformations of the wing, which are particularly noticeable when transitioning from powered to depowered states and during turns. These measurements of structural deformation are useful for assessing specific aspects of the wing deformation but can induce errors if used for trajectory estimations. Furthermore, the high centripetal accelerations of the kite during turns can lead to increased noise and sensor drift (Hesse et al., 2018).

- Wind Vanes: In the analysed setups, wind vanes were mounted in the bridle line system, where determining their orientation relative to the wing is challenging due to deformation of both the wing and the bridle lines. Recent wind tunnel calibration tests showed a mean absolute error of 2–3°, providing an estimate of the best achievable accuracy under controlled conditions for this setup. However, in flight, additional noise is introduced by vibrations in the bridle lines, which can further affect the measurements (Oehler and Schmehl, 2019). An alternative is to mount the vanes below the KCU, which mitigates most of these issues. Ideally, the vanes should be integrated with an IMU to allow for accurate orientation measurements relative to the ground. During development phases, booms mounted to the leading edge of the kite have also been employed (Borobia-Moreno et al., 2021), but this approach is generally unsuitable for commercial applications due to the fragility of the installation.

- Pitot Tube: Although Pitot tubes typically achieve good accuracy, they require regular maintenance and calibration, and external factors like ice, insects, or pollution can cause clogging (Ezzeddine et al., 2019). Furthermore, when used to measure wind speed, small errors propagate into larger errors due to the wind's small contribution to the total apparent speed. Recent wind tunnel tests showed that the current setup, once correctly calibrated, can achieve an accuracy within 5% for angles of attack up to 30°, but performance degrades at higher angles. Finally, the mounting position is critical; if the sensor is not mounted at the centre of rotation of the kite, it will measure velocity induced by the kite yaw rate, further amplifying inaccuracies.

Overall, the sensors that are least susceptible to the intrinsic deformations of the soft kite and the high accelerations of the system, and thus more reliable, are the GPS (for position and velocity), the load cell (for tether force), and the tether reel-out

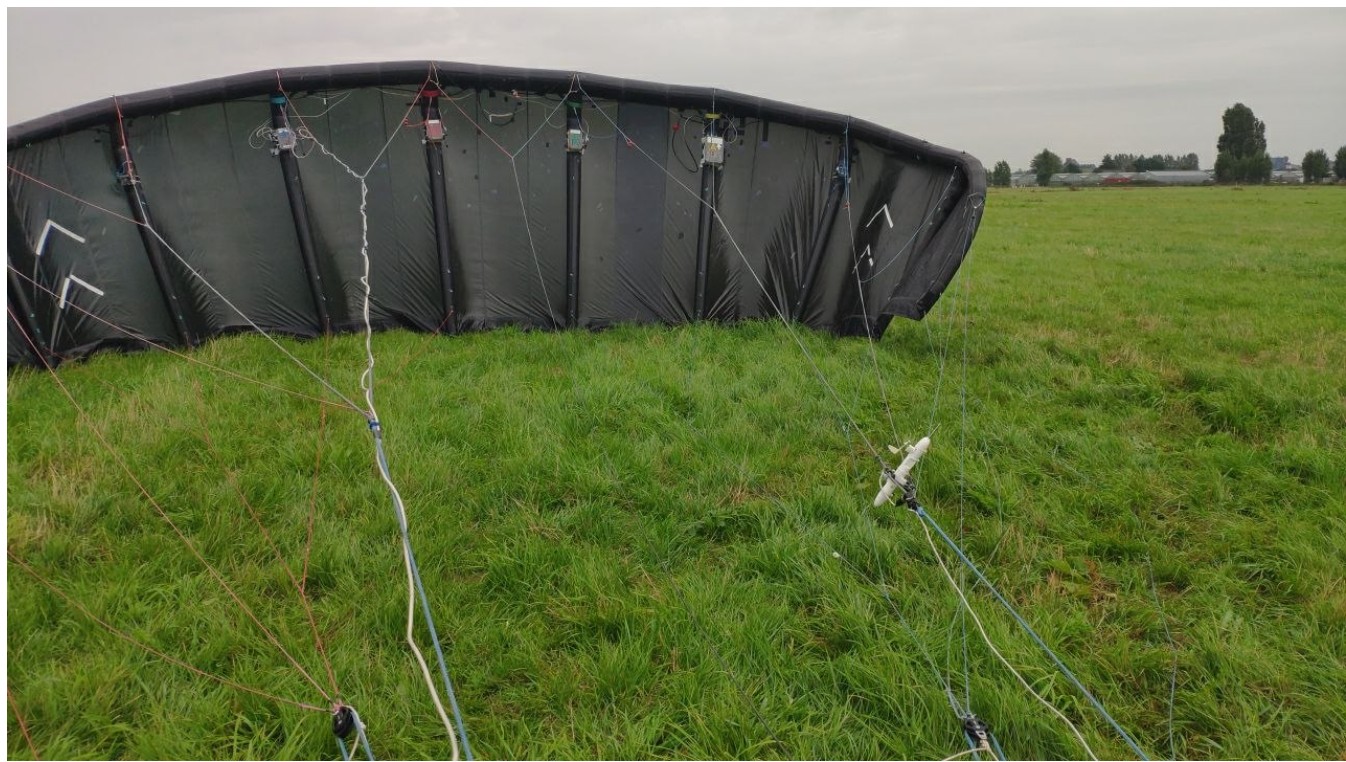

**Figure 3.** Fully instrumented V3.25 kite before launch. GPS+IMU units are visibly mounted in four of the inflatable struts. A Pitot tube and a wind vane are also installed in the bridle lines. Photo courtesy of Kitepower B.V.

encoder (for tether length and speed). These sensors can maintain their accuracy despite the flexible nature of the kite and are therefore used as the foundation of the sensor fusion model. This results in a minimal sensor setup sufficient for estimating kite motion. However, when the KCU is included in the model, an additional acceleration measurement—either of the KCU or the wing—is required to resolve its inertial effects. In this study, due to data availability, acceleration was obtained by numerically differentiating velocity measurements from the GPS. Further measurements, such as tether angles or airflow data, are optional

and can potentially enhance the accuracy of the estimations.

## 4 Filter design

In this section, a filter is designed to estimate the state of a tethered flying system by integrating a dynamic model and an observation model in an EKF. All vectors are expressed in the local East-North-Up (ENU) coordinate frame, with the exception of the Euler angles, which are computed in the North-East-Down (NED) frame to avoid discontinuities and jumps in their

representation. The dynamic model simulates the kite translational motion governed by aerodynamic, gravitational, and elastic tether forces. The tether dynamics are represented using a quasi-static lumped mass approach proposed by Williams (2017),

expanded to include the KCU's inertial and gravitational effects (Schelbergen et al., 2024). The observation model is constructed based on the availability of reliable measurements. As a minimum, the EKF requires the kite's position and velocity, tether force, and reel-out speed. If the KCU is included in the model, a measurement of the wing or KCU acceleration is also required. Additional optional measurements (e.g., tether angles or airflow) can be incorporated to refine the estimates.

## 4.1 Dynamic model

The dynamic model represents the kite wing as a point mass ($m_\mathrm{k}$) following Newton's second law. Its acceleration results from the sum of the tether force at the kite $\mathbf{F}_\mathrm{t,k}$, aerodynamic force $\mathbf{F}_\mathrm{a,k}$, and weight $\mathbf{F}_\mathrm{g,k}$. The components of the aerodynamic force are expressed as a function of the kite apparent velocity $\mathbf{v}_\mathrm{a}$ and the vector of aerodynamic coefficients $\mathbf{C}_\mathrm{a} = (C_\mathrm{L}, C_\mathrm{D}, C_\mathrm{S})$, which are assumed to be time-invariant,

$$
\mathbf{F}_\mathrm{a,k} = \begin{cases}
\mathbf{F}_\mathrm{L} = \frac{1}{2}\rho A_\mathrm{k} C_\mathrm{L} \|\mathbf{v}_\mathrm{a}\|^2 \mathbf{e}_\mathrm{L}, \\
\mathbf{F}_\mathrm{D} = \frac{1}{2}\rho A_\mathrm{k} C_\mathrm{D} \|\mathbf{v}_\mathrm{a}\|^2 \mathbf{e}_\mathrm{D}, \\
\mathbf{F}_\mathrm{S} = \frac{1}{2}\rho A_\mathrm{k} C_\mathrm{S} \|\mathbf{v}_\mathrm{a}\|^2 \mathbf{e}_\mathrm{S},
\end{cases}
\tag{4}
$$

where $\rho$ is the air density, $A_\mathrm{k}$ is the projected area of the wing in the plane defined by the two central struts, $C_\mathrm{L}$, $C_\mathrm{D}$, and $C_\mathrm{S}$ are the lift, drag, and side-force coefficients, respectively, and $\mathbf{e}_\mathrm{L}$, $\mathbf{e}_\mathrm{D}$ and $\mathbf{e}_\mathrm{S}$ are unit vectors in the directions of the lift force, drag force and side-force, respectively. The drag force acts in the direction of $\mathbf{v}_\mathrm{a}$, the lift force acts in the opposite direction of the tether force projected in the perpendicular plane to $\mathbf{v}_\mathrm{a}$, and the side force acts orthogonally to both. The apparent velocity is a function of the kite velocity $\mathbf{v}_\mathrm{k}$ and the wind velocity $\mathbf{v}_\mathrm{w}$, which is also assumed to be time-invariant,

$$
\mathbf{v}_\mathrm{a} = \mathbf{v}_\mathrm{w} - \mathbf{v}_\mathrm{k}.
\tag{5}
$$

Although the dynamic model assumes constant wind velocity and time-invariant aerodynamic coefficients, these quantities are treated as estimated states in the EKF. Their variability over time is captured by introducing appropriate process noise in the filter. In this sense, the coefficients and wind velocity are not fixed but are allowed to evolve during the estimation process to best fit the measured dynamics. The wind velocity can be accounted for using two different approaches: firstly, assuming it is time-invariant and not dependent on height, and secondly, using a logarithmic relation with height for its horizontal component. The latter can be done by means of the friction velocity $u_*$ and wind direction $\phi_\mathrm{w}$ instead of the horizontal wind components such that the height-dependent horizontal wind speed $v_\mathrm{w,h}$ is given by (Watson, 2023a),

$$
v_\mathrm{w,h} = \frac{u_*}{\kappa} \log \frac{z}{z_0},
\tag{6}
$$

where $z$ is the height above the ground, $\kappa \approx 0.4$ is the von Karman constant and $z_0$ is the surface roughness length. Even though this approach generally improves estimations by incorporating physical knowledge of the wind profile, it can lead to poorer estimations compared to modelling it as height-independent if the actual wind profile deviates from the assumed logarithmic model.

The tether force at the kite is determined by assuming a shape derived from a quasi-static force equilibrium, detailed in Williams (2017), with the addition of accounting for the KCU and its localised mass, representing the kite by two separate point masses (Schelbergen and Schmehl, 2024). The tether model uses a lumped mass approach with point masses connected

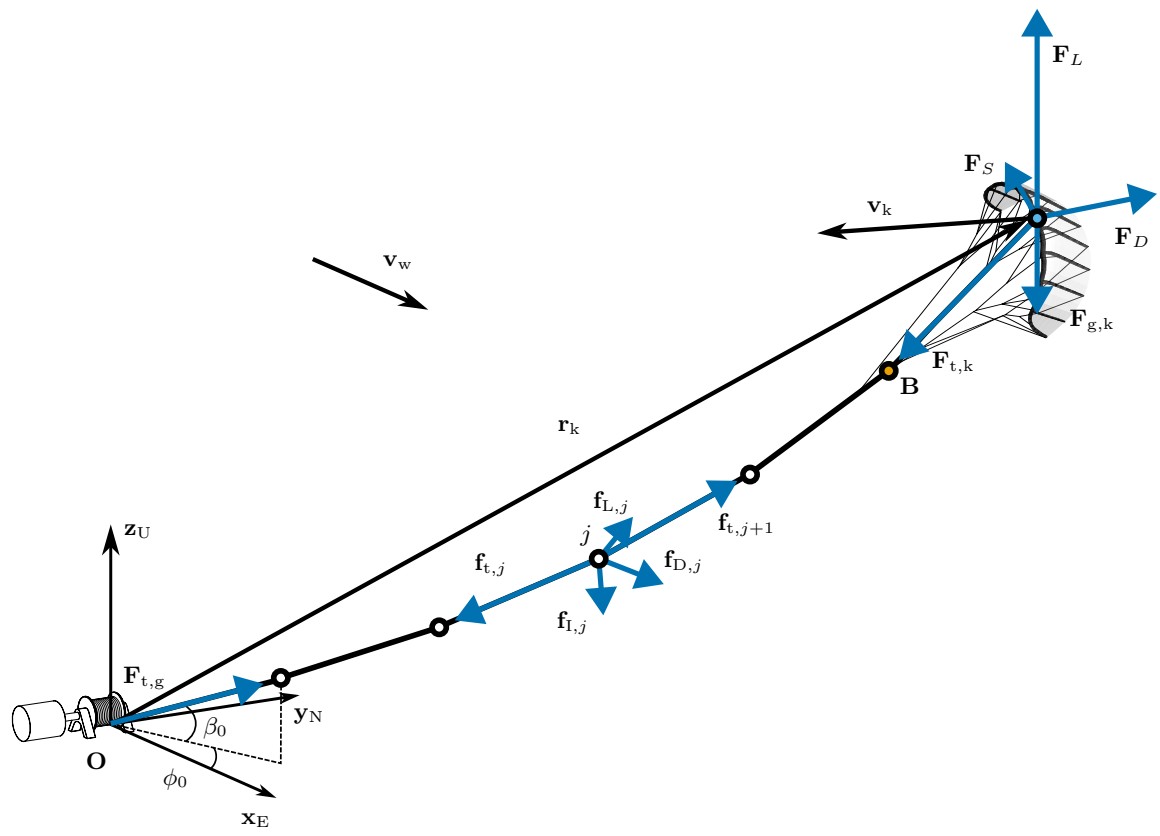

**Figure 4.** Schematic representation of the tether model and kite forces, as modelled by the dynamic system in the EKF. Adapted from Schelbergen and Schmehl (2024).

by spring elements (see Fig. 4). The shape of the tether is calculated based on the tether force at the ground $F_{t,g}$, the position $\mathbf{r}_k$ velocity $\mathbf{v}_k$ and acceleration $\mathbf{a}_k$ of the kite wing, the wind velocity, the azimuth $\phi_0$ and elevation $\beta_0$ of the first tether segment,

and the total deployed tether length $l$, all of which are either incorporated into the dynamic model or directly measured and used as inputs.

     The tether force at the wing is calculated using a shooting method. In this approach, the direction of each subsequent tether segment is determined by the sum of the elastic, drag, gravitational, and inertial forces. The method requires an initial estimate of the tether length as well as the magnitude and orientation of the tether force at the ground. Inertial forces for each segment

are determined by their centripetal acceleration, with the segment lengths $l_\text{i}$ adding up to the total tether length, including the bridle segment.

To compute these inertial forces, the velocities $\mathbf{v}_j$ and accelerations $\mathbf{a}_j$ of the discrete point masses along the tether are estimated under the assumption that they all rotate with a fixed angular velocity $\boldsymbol{\omega}$ (Williams, 2017), behaving like particles of a rigid body (see Fig. 5) . The velocity at the tether attachment point $\mathbf{B}$ particle can be represented assuming a purely rotational

motion about a point, the instantaneous centre of rotation (Meriam et al., 2018), which is unique for each tether particle.

$$\mathbf{v}_\text{B} = \boldsymbol{\omega} \times \mathbf{r}_{\text{c,B}}, \tag{7}$$

where $\mathbf{r}_{\text{c},B}$ is the vector from $\mathbf{B}$ to its instantaneous centre of rotation, which is perpendicular to $\boldsymbol{\omega}$. The acceleration can then

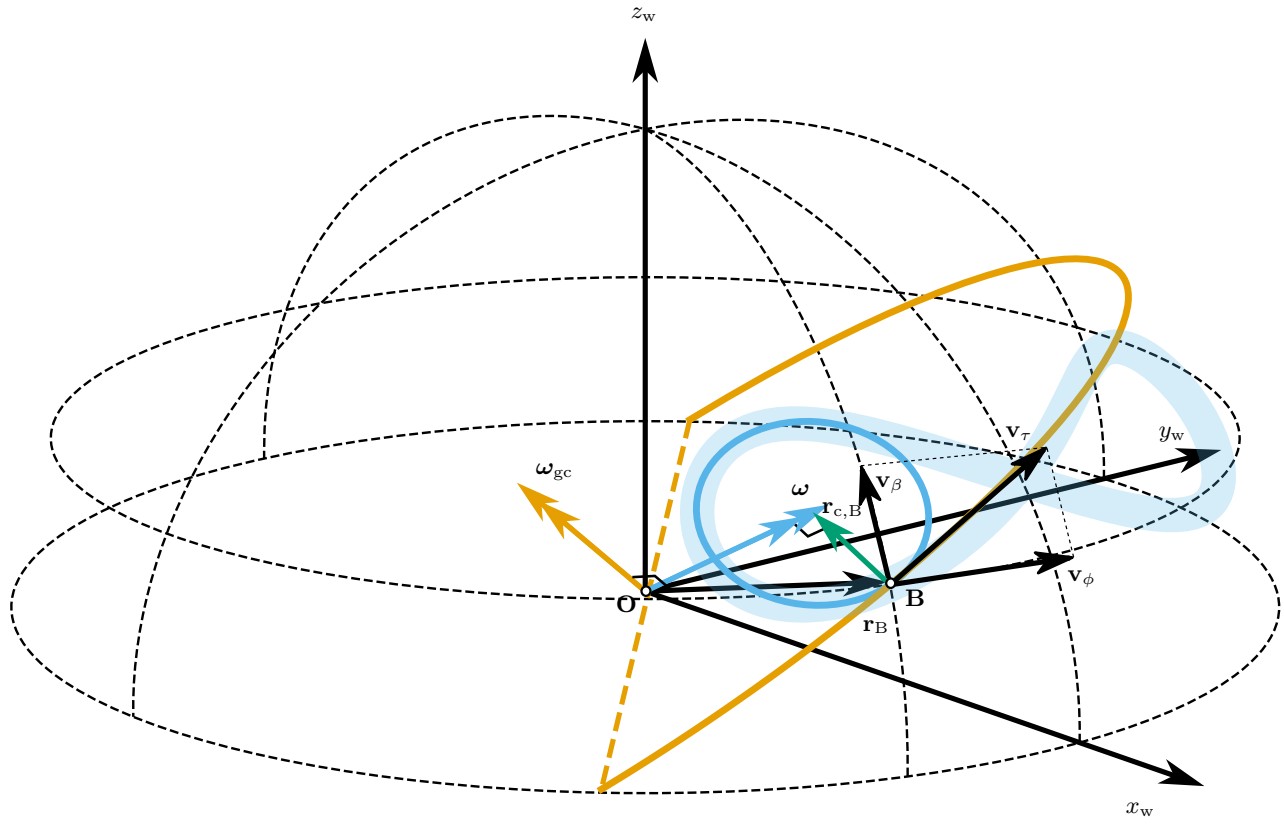

**Figure 5.** The angular velocity during straight flight, $\boldsymbol{\omega}_{\text{gc}}$ (orange), and during turns, $\boldsymbol{\omega}$ (blue), for a kite linked by a straight and inelastic tether. Adapted from Schelbergen and Schmehl (2024).

be estimated by differentiating the velocity,

$$\mathbf{a}_\text{B} = \frac{d\mathbf{v}_\text{B}}{dt} = \frac{d(\boldsymbol{\omega} \times \mathbf{r}_{\text{c,B}})}{dt} = \frac{d\boldsymbol{\omega}}{dt} \times \mathbf{r}_{\text{c,B}} + \boldsymbol{\omega} \times \frac{d\mathbf{r}_{\text{c,B}}}{dt}, \tag{8}$$

which can be expressed as,

$$\mathbf{a}_B = \boldsymbol{\alpha} \times \mathbf{r}_{c,B} + \boldsymbol{\omega} \times \mathbf{v}_B. \tag{9}$$

Here, $\boldsymbol{\alpha} = d\boldsymbol{\omega}/dt$ is the angular acceleration. The two terms of the acceleration represent the tangential acceleration $\mathbf{a}_\tau = \boldsymbol{\alpha} \times \mathbf{r}_{c,B}$ and the centripetal acceleration $\mathbf{a}_n = \boldsymbol{\omega} \times \mathbf{v}_B$.

If the acceleration and velocity of the tether attachment point $\mathbf{B}$ at the kite are known, we can compute the angular velocity,
instantaneous centre of rotation, and angular acceleration as follows, assuming the tangential acceleration to be in the direction of the kinematic velocity,

$$\boldsymbol{\omega} = \frac{\mathbf{v}_B \times \mathbf{a}_n}{\|\mathbf{v}_B\|^2}, \quad \mathbf{r}_{c,B} = \frac{\boldsymbol{\omega} \times \mathbf{v}_B}{\|\boldsymbol{\omega}\|^2}, \quad \boldsymbol{\alpha} = \frac{\mathbf{r}_{c,B} \times \mathbf{a}_\tau}{\|\mathbf{r}_{c,B}\|^2}. \tag{10}$$

On the straight flight path segments, the kite moves on a great circle trajectory, and $\boldsymbol{\omega}_{gc}$ is perpendicular to the tether, resulting in the centre of rotation being located at the ground station (Schelbergen and Schmehl, 2024). However, during turns, the centre
of rotation aligns with $\boldsymbol{\omega}$, and at each point along the tether, the centre of rotation will lie in the plane perpendicular to the angular velocity vector (see Fig. 5).

Given the angular velocity $\boldsymbol{\omega}$ and acceleration $\boldsymbol{\alpha}$, the velocity and acceleration at an arbitrary point 2 on the rigid body can be determined relative to a reference point 1 as follows,

$$\mathbf{v}_2 = \mathbf{v}_1 + \boldsymbol{\omega} \times \mathbf{r}_{2-1}, \tag{11}$$

$$\mathbf{a}_2 = \mathbf{a}_1 + \boldsymbol{\alpha} \times \mathbf{r}_{2-1} + \boldsymbol{\omega} \times (\boldsymbol{\omega} \times \mathbf{r}_{2-1}), \tag{12}$$

where $\mathbf{r}_{2-1}$ is the position vector from point 2 to 1. The velocity at the ground tether attachment point is assumed to be zero by neglecting the reel-in or reel-out motion. While this reeling motion can induce non-negligible velocities, particularly in the lower tether segments, we consider its effect negligible for the computation of tether drag, which primarily depends on the relative motion of the tether segments through the air. Consequently, the velocities and accelerations along the tether, relative
to the ground point, can be written as,

$$\mathbf{v}_j = \boldsymbol{\omega} \times \mathbf{r}_j, \tag{13}$$

$$\mathbf{a}_j = \boldsymbol{\alpha} \times \mathbf{r}_j + \boldsymbol{\omega} \times (\boldsymbol{\omega} \times \mathbf{r}_j). \tag{14}$$

In practice, assuming the kite-tether assembly to be a rigid body can lead to inaccuracies. Firstly, the kite can rotate freely around $\mathbf{B}$[1], and secondly, the tether deforms and sags. As a result, the angular velocity vector $\boldsymbol{\omega}$ is no longer aligned with the
280 ideal rotation axis and does not pass through the ground station.

Initial attempts to model the assembly as a single rigid body yielded limited success, leading to the decision to treat the kite and tether as two independent rigid bodies. This approach significantly improved the estimations.

---

[1]In this work, we intentionally collocate the bridle point with the KCU for modelling simplicity. However, this was not the case for the V3 kite, as shown in Figs. 1 and 4, where the bridle point is slightly lower than the KCU.

Since the kinematic measurements are obtained at the kite wing, the angular velocity of the kite, $\boldsymbol{\omega_k}$, is calculated using the accelerations and velocities at the wing,

$$\boldsymbol{\omega_k} = \frac{\mathbf{v}_k \times \mathbf{a}_{n,k}}{\|\mathbf{v}_k\|^2}, \quad \mathbf{r}_{c,k} = \frac{\boldsymbol{\omega_k} \times \mathbf{v}_k}{\|\boldsymbol{\omega_k}\|^2}, \quad \boldsymbol{\alpha} = \frac{\mathbf{r}_{c,k} \times \mathbf{a}_\tau}{\|\mathbf{r}_{c,k}\|^2}. \tag{15}$$

Subsequently, the velocities and accelerations are translated to the bridle point $\mathbf{B}$, representing the KCU, using Eq. (11).

Regarding the tether, since the position at $\mathbf{B}$ is not measured, the kite position is used to estimate its angular velocity using Eq. (13), under the assumption that it performs a great circle rotation around the ground station (Williams, 2017),

$$\boldsymbol{\omega_t} = \frac{\mathbf{r}_k \times \mathbf{v}_k}{\|\mathbf{r}_k\|^2}. \tag{16}$$

This assumption introduces inaccuracies, particularly in the acceleration values of each tether segment. However, since the point masses of the tether are relatively small, this does not significantly impact the overall accuracy of the model, as shown in Williams (2017). Assuming a constant angular velocity ($\boldsymbol{\alpha} \approx 0$), the velocities and accelerations along the tether are calculated as (Eq. (13)) (Williams, 2017),

$$\mathbf{v}_j = \boldsymbol{\omega_t} \times \mathbf{r}_j, \tag{17}$$

$$\mathbf{a}_j = \boldsymbol{\omega_t} \times (\boldsymbol{\omega_t} \times \mathbf{r}_j). \tag{18}$$

The tangential velocity at the bridle point is then given by,

$$\mathbf{v}_\tau = \boldsymbol{\omega_t} \times \mathbf{r}_\mathbf{B}. \tag{19}$$

This tangential velocity vector is projected into the horizontal and vertical planes, yielding $v_\phi$ and $v_\beta$, to estimate the rate of change of the tether orientation angles.

The aerodynamic force acting on the tether is estimated based on the cross-flow principle, where the flow components parallel and perpendicular to the body are treated independently (Hoerner, 1965; Bootle, 1971), which was found to have a good relation with test data for sub-critical flows ($Re_{\mathrm{crit}} \approx 3.5 \times 10^5$), where the Reynolds number is formed using the tether diameter as the characteristic length. The lift $\mathbf{f}_{L,j}$ and drag $\mathbf{f}_{D,j}$ forces acting on each tether segment can then be estimated as follows (Dunker, 2018),

$$\mathbf{f}_{L,j} = \frac{1}{2}\rho \left( C_\perp \sin^2 \alpha_j \cos \alpha_j - \pi C_\| \cos^2 \alpha_j \sin \alpha_j \right) l_j d_t v_{a,j}^2 \mathbf{e}_{L,j}, \tag{20}$$

$$\mathbf{f}_{D,j} = \frac{1}{2}\rho \left( C_\perp \sin^3 \alpha_j + \pi C_\| \cos^3 \alpha_j \right) l_j d_t v_{a,j}^2 \mathbf{e}_{D,j}, \tag{21}$$

where $C_\perp$ is the drag coefficient in the direction perpendicular to the tether, $C_\|$ is the skin friction drag coefficient (along the tether), $\alpha_j$ is angle of attack of the tether segment (which is $90°$ when the flow is perpendicular to the tether), $l_j$ is the length of the tether segment, and $d_t$ is the tether diameter. The direction of the drag force aligns with the apparent velocity of the segment, while the lift is directed perpendicular to the drag and lies in the plane defined by the apparent velocity and the tether direction, where $\mathbf{e}_{L,j}$ and $\mathbf{e}_{D,j}$ are unit vectors in the direction of these forces.

Similarly, the aerodynamic forces acting on the KCU are estimated by simplifying its geometry to that of a cylinder and applying the cross-flow principle, using experimentally derived coefficients for cylinders with different aspect ratios in the normal and tangential directions (Blevins, 1984), assuming the KCU is pitched 90° relative to the tether. However, since the KCU operates in a supercritical flow regime, the cross-flow principle may not provide an accurate approximation. A more suitable model for this regime is beyond the scope of the current project and will be considered in future work. Despite this, the contribution of the KCU drag to the overall system is very small compared to the other forces acting on the KCU and the system, so any inaccuracy here will not have a significant effect. This is further illustrated by the analysis presented in Sect. 5.

The EKF state vector comprises the kite wing position and velocity, aerodynamic coefficients, wind velocity, tether length, and azimuth and elevation of the first tether segment. The tether length is determined by the reel-out speed ($v_t$), while the orientation of the first tether segment depends on its angular velocity relative to the ground station, which can be approximated with the tangential velocity and radial distance of the tether attachment point. The tether force at the ground $F_{t,g}$ is measured using a load cell and given as an input. With these considerations, the dynamics of the model can be written as a function of the state vector ($\mathbf{x}$) and the input vector ($\mathbf{u}$),

$$\mathbf{x} = (\mathbf{r}_k, \mathbf{v}_k, \mathbf{v}_w, C_L, C_D, C_S, l_t, \beta_0, \phi_0), \qquad \mathbf{u} = (v_t, F_{t,g}). \tag{22}$$

The full system of ordinary differential equations (ODEs) to be solved is,

$$\mathbf{f}(\mathbf{x}, \mathbf{u}) = \begin{cases} \dot{\mathbf{r}}_k = \mathbf{v}_k & \text{(23a)} \\[2mm] \dot{\mathbf{v}}_k = \dfrac{\mathbf{F}_{t,k}\left(\mathbf{r}_k, \mathbf{v}_k, \mathbf{v}_w, l_t, \beta_0, \phi_0, F_{t,g}\right) + \mathbf{F}_{a,k}(\mathbf{C}_a, \mathbf{v}_w, \mathbf{v}_k) + \mathbf{F}_{g,k}}{m_k} & \text{(23b)} \\[3mm] (\dot{\mathbf{v}}_w = 0) \quad \text{or} \quad (\dot{u}_* = 0, \quad \dot{\phi}_w = 0, \quad v_{w,z} = 0) & \text{(23c)} \\[2mm] \dot{C}_L = 0, \quad \dot{C}_D = 0, \quad \dot{C}_S = 0 & \text{(23d)} \\[2mm] \dot{l} = v_t & \text{(23e)} \\[2mm] \dot{\beta}_0 = \dfrac{v_\beta}{\|\mathbf{r}_k\|} & \text{(23f)} \\[3mm] \dot{\phi}_0 = \dfrac{v_\phi}{\|\mathbf{r}_k\|}, & \text{(23g)} \end{cases}$$

Additionally, it is possible to estimate a bias or offset $\delta$ in any the measurements that are directly modelled, such as the tether length and angles, by adding them as a time-invariant variable ($\dot{\delta} = 0$).

This model forms the basis for the EKF design, capturing the essential dynamics of the kite and tether system. The model can be expanded to fly-gen systems by accounting for an additional thrust force acting on the kite, which can be modelled as a function of the control inputs and the kite velocity. When added to Eq. (23b), the thrust can be an input or a state variable, depending on the EKF design.

## 4.2  Observation model

The required minimum measurements are the position and velocity of the kite wing. An additional observation is required to ensure that the position of the wing calculated with the tether model, $\mathbf{r}_{k,t}$, matches the estimated position of the kite wing, $\mathbf{r}_k$. This ensures an agreement between the position of the kite (Equation 23a) and the shape of the tether (defined by the tether length and the orientation of the first tether segment). Therefore, the observation model vector is given by,

$$\mathbf{h}(\mathbf{x},\mathbf{u}) = \begin{cases} \mathbf{r}_{k,m} = \mathbf{r}_k + \eta_{\mathbf{r}_k} & \text{(24a)} \\[6pt] \mathbf{v}_{k,m} = \mathbf{v}_k + \eta_{\mathbf{v}_k} & \text{(24b)} \\[6pt] \mathbf{0} = (\mathbf{r}_k - \mathbf{r}_{k,t}) & \text{(24c)} \\[6pt] \text{—optional—} & \\[6pt] v_{a,m} = \|\mathbf{v}_w - \mathbf{v}_k\| + \delta_{v_a} + \eta_{v_a} & \\[6pt] l_m = l + \delta_l + \eta_l & \\[2pt] & \text{(24d)} \\[2pt] \beta_{0,m} = \theta_0 + \delta_{\beta_0} + \eta_{\beta_0} & \\[6pt] \phi_{0,m} = \phi_0 + \delta_{\phi_0} + \eta_{\phi_0} & \\[6pt] \alpha_m = \alpha + \delta_\alpha + \eta_\alpha, & \end{cases}$$

where the subscript m refers to the measured quantities, $\eta_i$ represents the Gaussian-distributed measurement noise and $\delta_i$ the measurement offset for each variable $i$.

Additionally, several other measurements can be incorporated into the EKF to enhance its accuracy, such as the tether length, the orientation of the tether measured at the ground, the apparent wind speed, and the angle of attack, each potentially subject to an offset. However, it is important to recognize that poorly calibrated or faulty equipment can introduce significant errors in the estimation process.

### 4.2.1  Sensor offset correction

The EKF is designed to correct sensor biases, such as those in the tether length and angle measurements, by incorporating their offset as a state within the filter. This allows the filter to estimate and subtract the offset from sensor readings.

However, if the measurement is not directly modelled as a state variable, the filter may fail to estimate its offset accurately, requiring alternative methods. This problem is particularly relevant for airflow sensors like Pitot tubes and wind vanes. In such cases, a correction procedure is advised by initializing the filter without using the biased sensors. The estimated states are then used to calibrate the sensors after the filter has converged, provided the filter has been correctly pre-tuned to the system at hand.

Some measurements, such as the Euler angles from the IMU, are not directly modelled in the filter, and their offsets are corrected using the orientation of the bridle segment, which closely follows the measured orientation of the wing. However, if the IMU is positioned at the wing, the sensor will also measure the deformations due to the actuation. Larger deformations around the central strut of the wing are observed during reel-in and can be estimated based on the linear relationship between

the variation in pitch and the depower setting $u_p$, which is directly linked to the length of the depower tape. By comparing the EKF-estimated pitch with the measured pitch $\theta_{\mathrm{m}}$, it is possible to infer the pitch sensor offset $\delta_\theta$ and the depower angle $\alpha_{\mathrm{d}}$. This comparison allows the extraction of the kite's rigid-body pitch, $\theta_{\mathrm{k}}$, from the measured sensor data, correcting for offset and deformation to match the orientation of the bridle segment in the tether model,

$$\theta_{\mathrm{k}} = \theta_{\mathrm{m}} - \delta_\theta - \alpha_{\mathrm{d}} \tag{25}$$

The yaw angle, however, can only be estimated directly by incorporating the yaw rate into the filter model. Nevertheless, the anhedral shape of the kite naturally aligns it with the apparent wind in average, enabling an offset correction based on this tendency. The good agreement between estimations and measurements, along with the offsets for the Euler angles mentioned earlier, is presented in Sect. 5.

## 4.3 Extended Kalman Filter

In this section, the process followed by the iterated EKF to correct the measured states and predict the unknowns is described. This is illustrated in the flowchart in Fig. 6, where the hatted symbols denote the predicted states, and the superscript $^*$ represents the nominal state around which the EKF is linearised. In the context of the iterated EKF, the nominal state corresponds to the current best estimate of the true state at each iteration, and it is updated during the measurement update step to improve the accuracy of the linearisation. The algorithm follows the standard setup of an iterated EKF (Gibbs, 2011) with a slight modification. After the one-step-ahead prediction, where the dynamic model is propagated to the next timestep, the Jacobians of the observation and dynamic model vectors are calculated. In this step, the tether force at the kite, obtained with the tether model, is differentiated only with respect to the tether length and the first tether segment orientation, whilst the rest of the states it depends on are taken from the last predicted state, given as input in the Jacobian calculation. There are several reasons for this choice; first, the tether model (Williams, 2017), in its original formulation, solves an optimisation problem for these three variables, while all the other variables are assumed to be known. Second, and more importantly, the introduction of the wind and kite velocity within the tether model introduces so much non-linearity that the performance of the Kalman filter is degraded. Therefore, the exact dependency of the function $\mathbf{f}$, which defines the ODE of the system, on these states in the tether model is not accounted for when propagating the covariance matrices.

As shown in the remaining steps of the flowchart in Fig. 6, the iterated EKF follows a standard procedure to update its gain and state, as well as the covariance matrix of the state estimation error.

The EKF implementation was benchmarked for performance on a standard laptop. As detailed in Appendix C1, the filter runs over 50 times faster than real time, with low CPU and memory usage, demonstrating its suitability for real-time or embedded applications. The hardware and software specifications used for this benchmarking are listed in Appendix C2.

### 4.3.1 Tuning

One of the most crucial and complex aspects of a well-performing Kalman filter is tuning, which can be done by means of the state and measurement noise covariance matrices Q and R. For simplicity, these are defined as diagonal matrices, with the

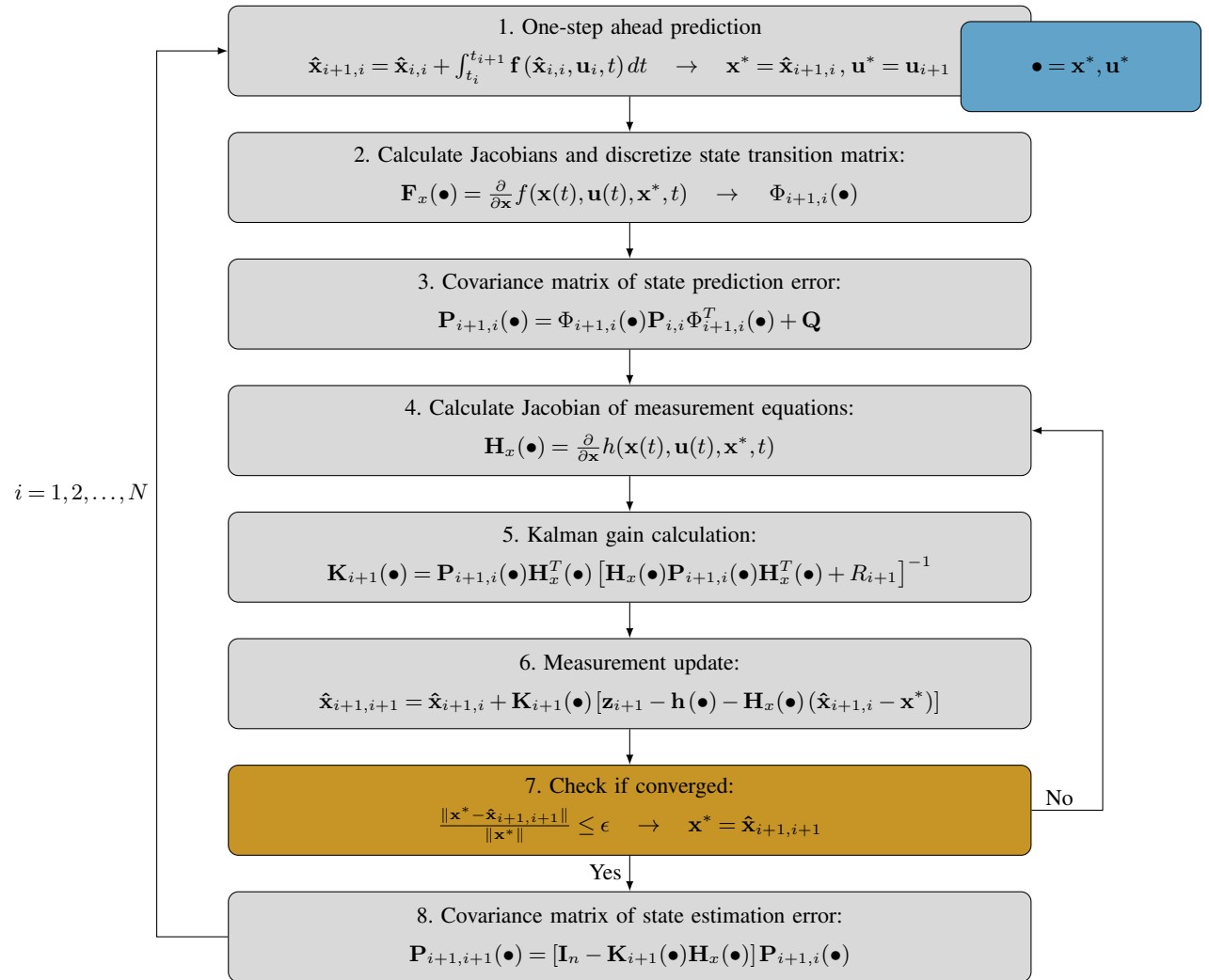

**Figure 6.** Iterated extended Kalman filter process flowchart.

diagonal elements representing the expected variance of the state and measurement noise, assuming a Gaussian-distributed noise with zero mean. These matrices were found to be system-dependent, meaning their optimal values can vary between different systems or kites. However, for the same kite, once calibrated, these coefficients maintain good performance even under varying environmental conditions.

The tuning of the EKF was performed manually, requiring an initial understanding of the magnitude and time dependence of the modelled parameters and a good knowledge of the accuracy of each sensor. There are, however, a few aspects that can be checked to ensure proper calibration. The first is to ensure that the wind speed and direction estimates do not show any pattern-related variations, such as periodic changes due to the figure-of-eight pattern flown by the kite. Moreover, the filter estimates for position and velocity should closely align with the measurements. Finally, the aerodynamic coefficients should remain within

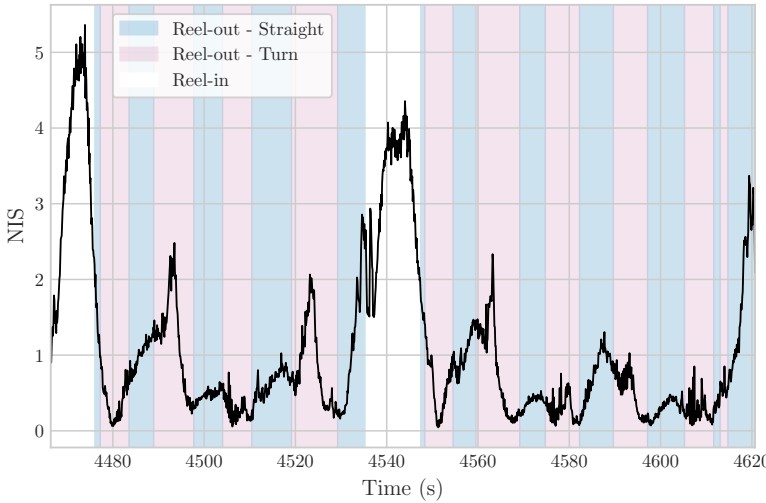

**Figure 7.** NIS metric for two power generation cycles.

the expected values of the flying wing. For a better understanding of the accuracy of the EKF estimates, analysing the time history of the filter performance parameters can be helpful. The normalized innovation squared (NIS) metric is commonly used for filter tuning in the absence of ground truth measurements (Bar-Shalom et al., 2002). It quantifies the consistency between the predicted measurements and the actual measurements, relative to the expected uncertainty.

$$\text{NIS} = \boldsymbol{\nu}_i^T \mathbf{S}_i^{-1} \boldsymbol{\nu}_i, \tag{26}$$

where $\boldsymbol{\nu}_i$ is the measurement residual and $\mathbf{S}_i$ the associated innovation covariance matrix.

As shown in Fig. 7, the performance decreases during turns and reel-in, which may indicate either that the measurements degrade during these phases or that the dynamic model cannot capture the relevant dynamics in these sections of the flight. Since position and velocity in the analysed datasets come from GPS+IMU-fused measurements, we believe the former to be the case, particularly during reel-in, as further discussed in the results section. For an optimally tuned filter, this metric should follow a chi-squared distribution; however, achieving this level of tuning is outside the scope of this work. For the leading-edge inflatable V3 kite, the standard deviations of the process and measurement noise terms detailed in Appendix B2 result in reasonable estimates.

## 5   Results

The datasets used in this study were acquired during three test flights conducted by Kitepower in the frame of two test campaigns. The first campaign took place in 2019 at the former naval air base Valkenburg, the Netherlands, using the 25 m$^2$ V3.25B kite developed by Kitepower on the basis of the TU Delft V3 kite (Poland and Schmehl, 2024). The selected dataset was published in Schelbergen et al. (2024) and analysed in Roullier (2020); Schelbergen and Schmehl (2024). It includes data from

**Table 2.** Overview of EKF models with corresponding sensor setups and wind model types.

| Model | Additional Measurements | Wind Model |
|-------|------------------------|------------|
| EKF 0 | - | Constant |
| EKF 1 | Tether length | Constant |
| EKF 2 | - | Logarithmic |
| EKF 3 | Apparent wind speed | Constant |
| EKF 4 | Zero vertical wind speed | Constant |
| EKF 5 | Tether length and angles | Constant |

two sensor boxes with GPS+IMU mounted on the two central struts of the wing, an airflow sensor comprising a Pitot tube and a single wind vane measuring the angle of attack, a load cell on the ground, and tether length and reeling speed sensors.

The second campaign took place 2023 to 2024 in Bangor Erris, Ireland, using the 60 m$^2$ V9 kite developed by Kitepower. This site in Northern Ireland is known for its consistently strong winds, predominantly from the south-west. The two selected flights of that campaign (Cayon et al., 2024a, b) include additional sensors used to study different measurement configurations. In the 2024 flight, two GPS+IMU units were mounted on the central struts of the wing, while in the other, one of the units was instead mounted on the KCU. Measurement data was complemented with profiling lidar readings recorded using a Windcube v2 (Vaisala), which is used to validate the wind estimations of the EKF.

This section evaluates various sensor setups and corresponding EKF configurations, as summarised in Table 2. Each model builds upon a baseline set of required measurements, which, as discussed in Sect. 3, includes the kite's position and velocity, tether force, reel-out speed, and the acceleration of the kite wing (to account for the inertial effects of the KCU). The additional measurements listed in Table 2, such as tether angles or apparent wind, are used to enhance estimation accuracy and assess the sensitivity of the filter to different sensor configurations.

The position and velocity data used in this study come from a GPS+IMU fused dataset processed with the embedded EKF of a Pixhawk sensor using PX4 autopilot. Unfortunately, only the filtered output of this onboard estimator was logged during the test flights, and raw GPS data were not recorded. It is acknowledged that using pre-filtered data introduces dynamics that may affect filter stability and consistency, particularly under tethered flight conditions for which the Pixhawk estimator was not designed.

## 5.1 Kite kinematics

One of the primary functions of the EKF is to enhance the accuracy of existing measurements, such as the kite position and velocity. By integrating additional information, the EKF refines data from standalone kinematic sensors like GPS. In Fig. 8, portions of two flights from the two campaigns are depicted in terms of azimuth, elevation, and radial distance. The azimuth is shown relative to the mean wind direction, with zero indicating alignment with the wind.

Overall, there is good agreement between the EKF estimations and the measurements throughout the flights. However, the highest discrepancies occur during the reel-in phase, when the kite is depowered, and to a lesser extent during turns. A significant factor contributing to these discrepancies is that the proprietary EKF of the Pixhawk relies on a model tailored to drones, which does not account for the constrained tethered flight dynamics of kites. This limitation likely contributes to inconsistencies in sensor readings, such as the mismatch between radial distance and tether length during reel-in, where the radial distance shows unphysical values several meters longer than the actual tether length. On the other hand, when the tether length is incorporated as an additional measurement, our EKF estimation aligns well with the tether length, correcting the position measurements to be consistent with the physical constraints of the system.

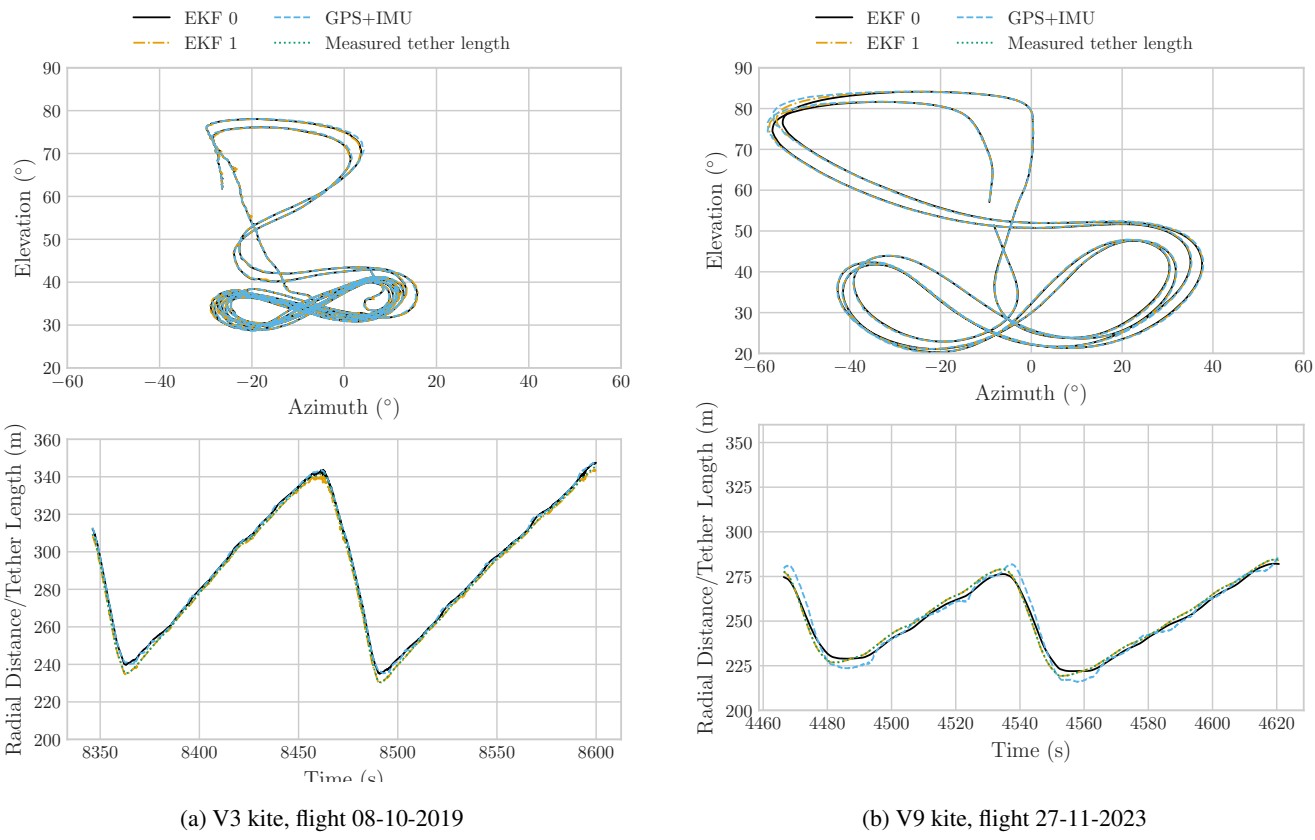

(a) V3 kite, flight 08-10-2019

(b) V9 kite, flight 27-11-2023

**Figure 8.** Comparison of estimated and measured kite trajectory, in terms of the azimuth and elevation angles and radial distance.

Figure 9 shows the kite and apparent wind speeds over two full flight cycles of the V3 kite. The kite is observed to speed up during turning manoeuvres and slow down while climbing along the straight segments of the figure-eight trajectory. This increase in speed is primarily driven by the kite's weight, as the reduced aerodynamic efficiency during turning, combined with the position further from the centre of the wind window, would otherwise result in a reduction in speed. Two EKF configurations are compared: EKF 3, which incorporates apparent wind speed as a measurement, and EKF 4, which instead

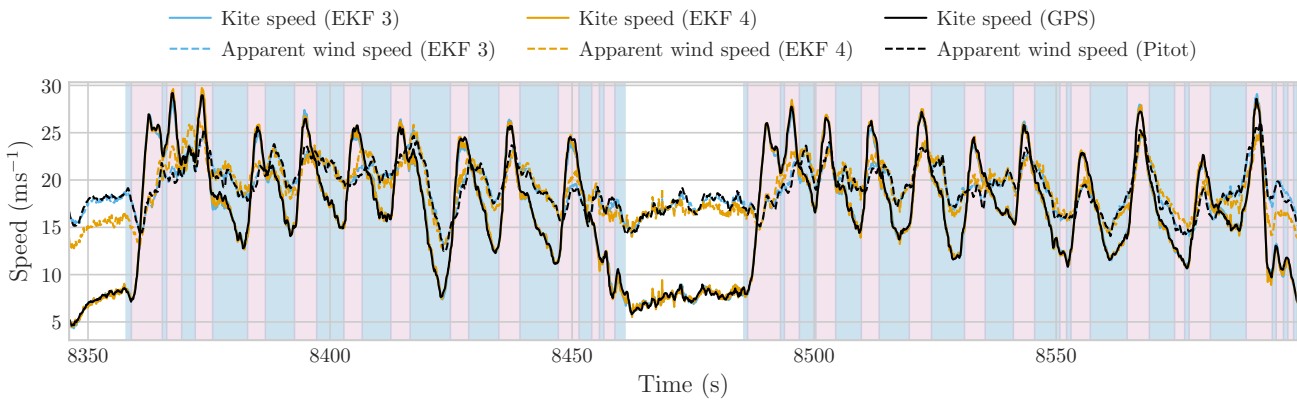

**Figure 9.** Comparison of the estimated and measured kite speed and apparent wind speed during the flight on 08-10-2019 using the V3 kite. The comparison is done for a configuration with (EKF 3) and without (EKF 4) apparent wind speed measurements.

imposes a constraint of zero vertical wind speed without using apparent wind measurements. While EKF 3 provides the best agreement with Pitot tube data, EKF 4 yields more consistent wind vector estimates than other configurations that similarly exclude apparent wind measurements but do not constrain the vertical component. This highlights the value of imposing
physical constraints when measurement data are limited. Apparent wind speed estimates from both EKFs align well with Pitot tube measurements, after applying an identified offset of $0.855 \text{ ms}^{-1}$. The apparent wind speed remains below the kite speed during turns, as the kite moves partially with the wind, and remains elevated during reel-in despite lower kite speeds, due to the kite flying into the wind.

In Fig. 10, the Euler angles estimated by the PX4 onboard EKF (based on IMU measurements) are compared with the
460 orientation of the bridle segment, defined by a pitch and a roll. The third Euler angle, the yaw, is not modelled by the EKF and has been computed by aligning the kite reference frame either with the apparent wind or the kinematic velocity directions. As summarised in Table 3, the results show a strong agreement between the tether model and the orientation measurements from the IMU. This level of consistency suggests that, for soft kites, a quasi-static two-point mass model—representing both the wing and the suspended KCU—can accurately capture the orientation of the kite.
Furthermore, although yaw is not directly modelled, there is a notable alignment between the estimated and measured angles when the kite is aligned with the apparent wind direction. Conversely, the yaw estimation error increases when aligned with the kite kinematic velocity. This behaviour suggests that the anhedral shape of the kite promotes a natural alignment with the local inflow. This is further evidenced by the sideslip angle measurements, available only in the V9 dataset, which show a standard deviation of approximately $2.5°$ around the mean, with peak values reaching up to $5°$ during turns.
By modelling the tether shape, which includes the KCU, the kite deformation at the IMU can be estimated by comparing the predicted orientation of the bridle segment from the EKF with the measured orientations at the wing. This approach allows for an approximate assessment of wing deformation during flight and helps isolate the rigid body orientation. Evidence of

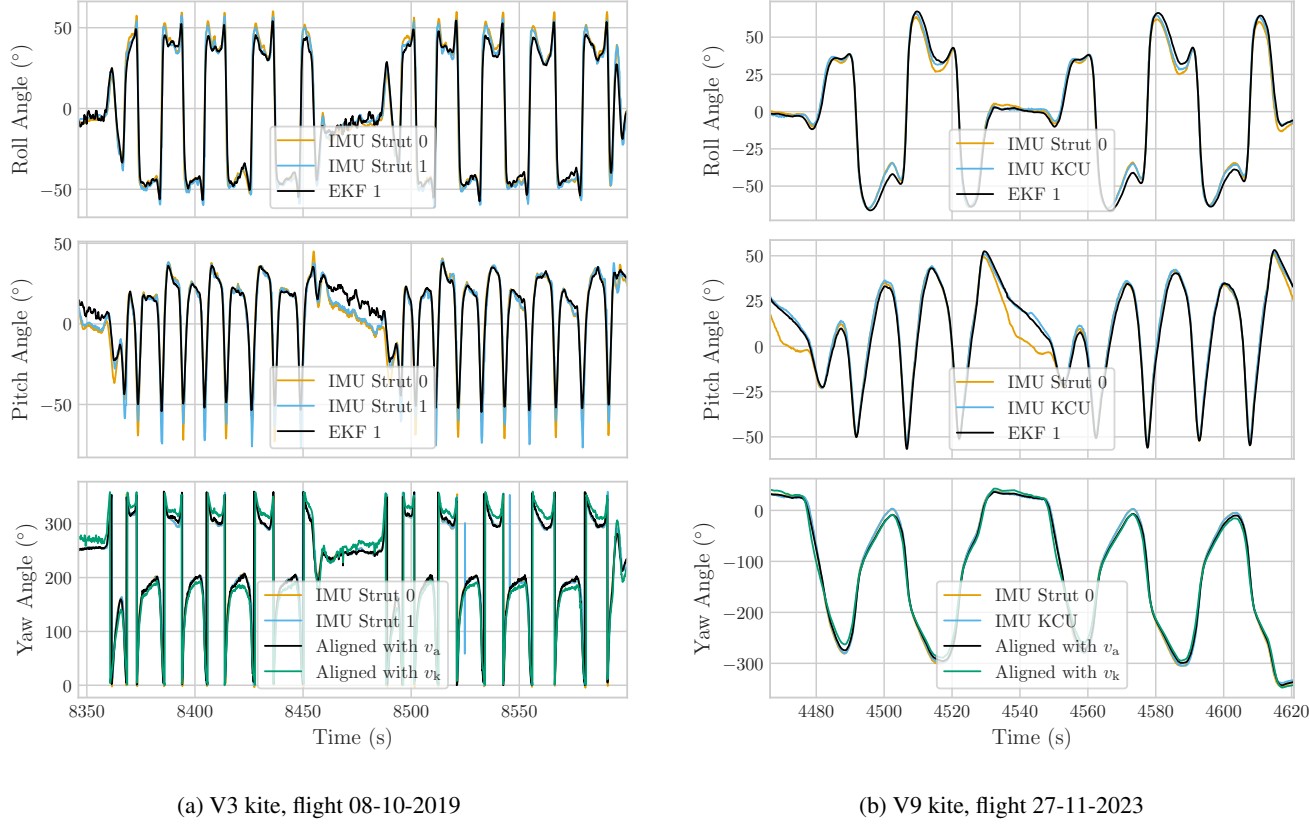

(a) V3 kite, flight 08-10-2019

(b) V9 kite, flight 27-11-2023

**Figure 10.** Comparison of estimated and measured Euler angles, with measurement biases removed using EKF estimations.

this deformation is visible during changes in the depower setting and in turning manoeuvres. Depower-induced deformation is evident in Fig. 10b, as indicated by the pitch offset observed during reel-in between the IMU on the central strut and both
the EKF estimate and the measurement at the KCU. A similar behaviour is observed in Fig. 10a, where the measured pitch deviates from the estimated value during reel-in. This information can be utilised to translate the measured angle of attack at the bridle to the wing angle of attack (see Eq. 1).

Regarding turning deformation, this effect is most pronounced in the V3 kite (see Fig. 10a), which corresponds to a kite whose structure exhibited greater overall deformation. Such deformation may be undesirable if the goal is to maintain aerody-
namic performance. However, this clear identification of turning deformation, which is more pronounced in one strut than the other depending on the turning direction, provides valuable insight into the aero-structural deformations of the wing (Schelbergen and Schmehl, 2024).

**Table 3.** Root mean squared errors (RMSE) of pitch, roll, and yaw for two kite models

| Orientation RMSE | V3 (08-10-2019) | V9 (27-11-2023) |
|---|---|---|
| Pitch | 3.44 deg | 3.07 deg |
| Roll | 3.90 deg | 2.91 deg |
| Yaw (aligned with $v_a$) | 3.83 deg | 4.65 deg |
| Yaw (aligned with $v_k$) | 14.66 deg | 11.73 deg |

## 5.2 System dynamics

Modelling the various components of the kite and tether system allows for the isolation of individual force components, enabling an assessment of their relative significance. Fig. 11 illustrates the different forces acting on the V3 kite system.

As expected, the lift force generated by the wing is the dominant contribution, primarily responsible for pulling the tether. It is followed by the wing drag force, which exhibits considerable variability, peaking during turns and decreasing during the reel-in phases. The parasitic drag of the tether and KCU can be seen to account for a relatively small portion of the total drag. The side force, though relatively low, plays a crucial role in balancing the forces during turns by providing the necessary lateral force to counteract the centripetal acceleration of the wing. The primary force component of the KCU is its inertia, which can be interpreted as an external force on the tether. In this particular system configuration, where the KCU was oversized relative to the wing, its inertia could reach up to 40% of the tether force during turns, exerting a greater influence than its weight due to the high accelerations during turns.

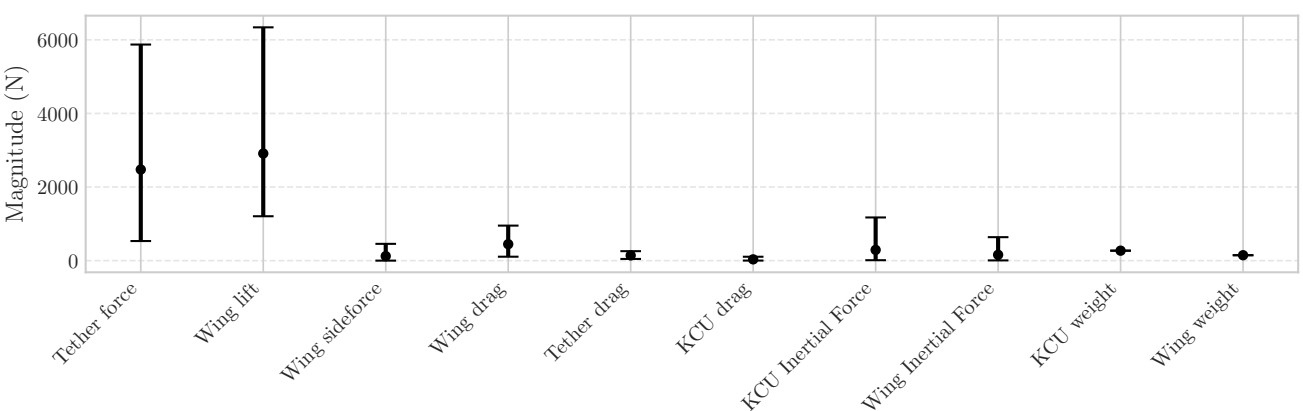

**Figure 11.** Mean, maximum, and minimum values of the different force contributions acting on the airborne subsystem during the flight on 08-10-2019 using the V3 kite. These forces were computed based on the estimated states obtained from the EKF described in Sect. 4.

Another novelty is that the current EKF can effectively estimate tether sag, defined as the difference between the tether length (i.e., the unstretched, deployed length) and the radial distance between the kite and the ground attachment point. As shown in Fig. 12, which presents the sag and ground tether force for two power generation cycles, there is a clear relationship between the two. As expected, the tether experiences the most sag during the reel-in phases, when the kite is depowered and the tether forces are lowest. Note that small negative sag values may occur due to elastic elongation of the tether under tension during the reel-out phase.

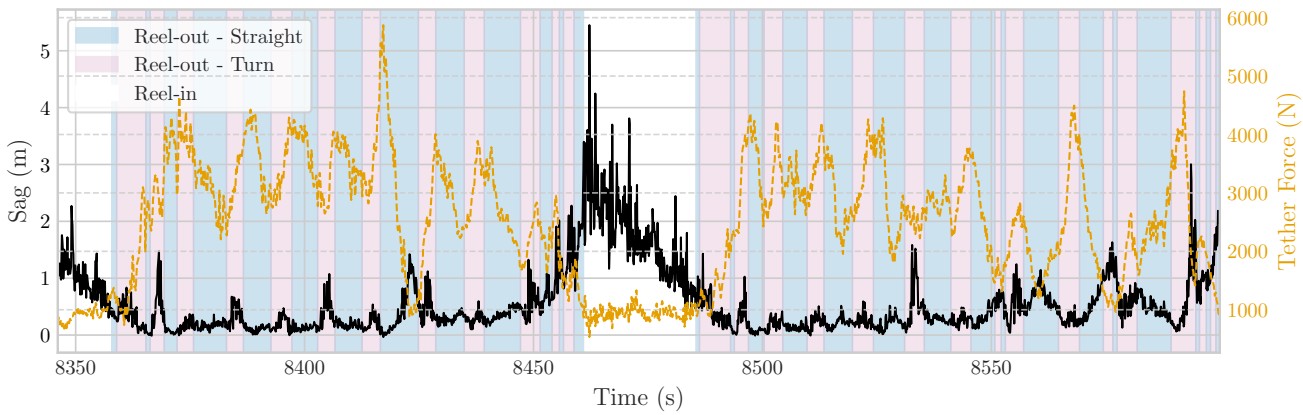

**Figure 12.** Variation of sag and tether force during two power generation cycles from the flight on 08-10-2019 using the V3 kite. The plot shows the sag in meters (left y-axis) and the tether force in Newtons (right y-axis).

Additionally, a consistent peak in sag is observed during turns, partly due to the reduced speed at the top of the figure-eight pattern (i.e., lower tether force). This effect is further amplified by the inertia of the KCU, which causes the kite to rotate relative to the tether. This misalignment between the aerodynamic forces and the tether force might lead the kite to dive into the sphere, causing the tether to sag.

### 5.2.1 Aerodynamic identification

To accurately estimate the aerodynamic performance of the kite, it is crucial to precisely estimate both the orientation of the tether and the wind speed and direction. The latter is particularly critical for determining the direction of the drag force and obtaining accurate estimates of the drag coefficient. Linking the instantaneous aerodynamic coefficients to the angle of attack at the wing $\alpha_\mathrm{w}$ adds another layer of complexity, as this angle is currently measured at the bridle. This section aims to improve the accuracy of these estimates by integrating deformation estimates from the EKF to calculate the angle of attack at the wing, thereby enhancing the prediction of the wing aerodynamic coefficients as a functions of the angle of attack.

Alternatively, the angle of attack can be calculated using the orientation of the wing measured by the IMU and the estimated apparent wind velocity. However, the accuracy of this method is limited by the time resolution quality of the wind velocity

estimates, making the measured angle of attack better at capturing temporal variations. Nevertheless, this angle is used to find $\alpha_{0,\mathrm{d}}$, which is subtracted from the measured angle at the bridle (see Eqs. (1, 2)),

In Fig. 8a, the measured trajectory of two power generation cycles is presented alongside the EKF estimates, with the azimuth angle centered on the mean wind direction. During this flight segment, the trajectory was slightly misaligned with the wind direction by approximately $10°$, according to the EKF predictions.

    In Fig. 13, the aerodynamic coefficients and angles of attack are plotted for the selected flight segment. The lift and drag coefficients remain relatively constant throughout the reel-out phase, with spikes during turns corresponding to decreased lift

and increased drag. This behaviour is consistent with observations reported in previous studies, such as Oehler and Schmehl (2019); Roullier (2020), where the increase in drag during turning manoeuvres was attributed to steering-induced deformation of the wing. Additionally, the side of the figure-eight pattern that is more misaligned with the wind direction shows a higher increase in drag coefficient and a smaller decrease in lift, while on the other side, the drag peaks are smaller, and the decrease in lift is higher. The increase in drag coefficient on the misaligned side of the figure of eight pattern could be attributed to a

higher sideslip angle, although relating this directly to the lift coefficient is less straightforward. The observed changes in lift might be linked to variations in the kite's trim angle, which could be influenced by shifts in aerodynamic polars with sideslip, although this relationship warrants further investigation.

    The parasitic drag of the tether, bridles and KCU contributes a significant portion of the total drag, approximately $30\%$ during reel-out and up to $50\%$ during reel-in. As for the angles of attack, the measured angle at the bridle lines remains

fairly constant throughout the flight, suggesting that the kite maintains pitch stability around a certain trim angle (Thedens and Schmehl, 2023; Cayon et al., 2023). The angle measured at the bridle lines can be translated to the wing angle of attack using the depower-induced deformations identified in Fig. 10a.

    The wing polars using this estimated angle are presented in Fig. 14 as mean values, with shaded areas indicating the $99\%$ confidence interval. The angle of attack shown corresponds to the wing angle of attack, obtained by translating the measured

angle of attack at the bridles to the wing reference frame, accounting for an offset $\alpha_{\mathrm{d},0}$ and the depower angle $\alpha_{\mathrm{d}}$ (see Eqs. 1, 2). As inferred from the angle of attack and aerodynamic coefficient estimates shown in Fig. 13, the kite exhibits relatively constant behaviour for a fixed depower setting, with standard deviations around the mean of $1.91°$ and $0.84°$ during reel-out and reel-in, respectively. This limited variability constrains the range of angles of attack explored during the flight. Despite this constraint, portions of the polar curves can still be estimated. However, for a complete aerodynamic characterisation of the

kite, tailored test flights should be conducted, where the kite is forced to dynamically change the angle of attack to explore the wider range of conditions.

    The experimentally derived polars are compared with findings from previous studies employing different levels of fidelity. The Reynolds-Averaged Navier–Stokes (RANS) simulations by Viré et al. (2022); Lebesque (2020) were conducted using the CAD model of the V3 kite at a Reynolds number of $3 \times 10^6$, whereas the results from the Vortex Step Method (VSM)

incorporate an aero-structural solver that accounts for kite deformation caused by actuation inputs (Cayon et al., 2023; Poland and Schmehl, 2023). Additionally, the estimations are compared with an experimental study of the same dataset that used a simpler tether model (Roullier, 2020). In this study, the wind speed at the kite was extrapolated from ground measurements

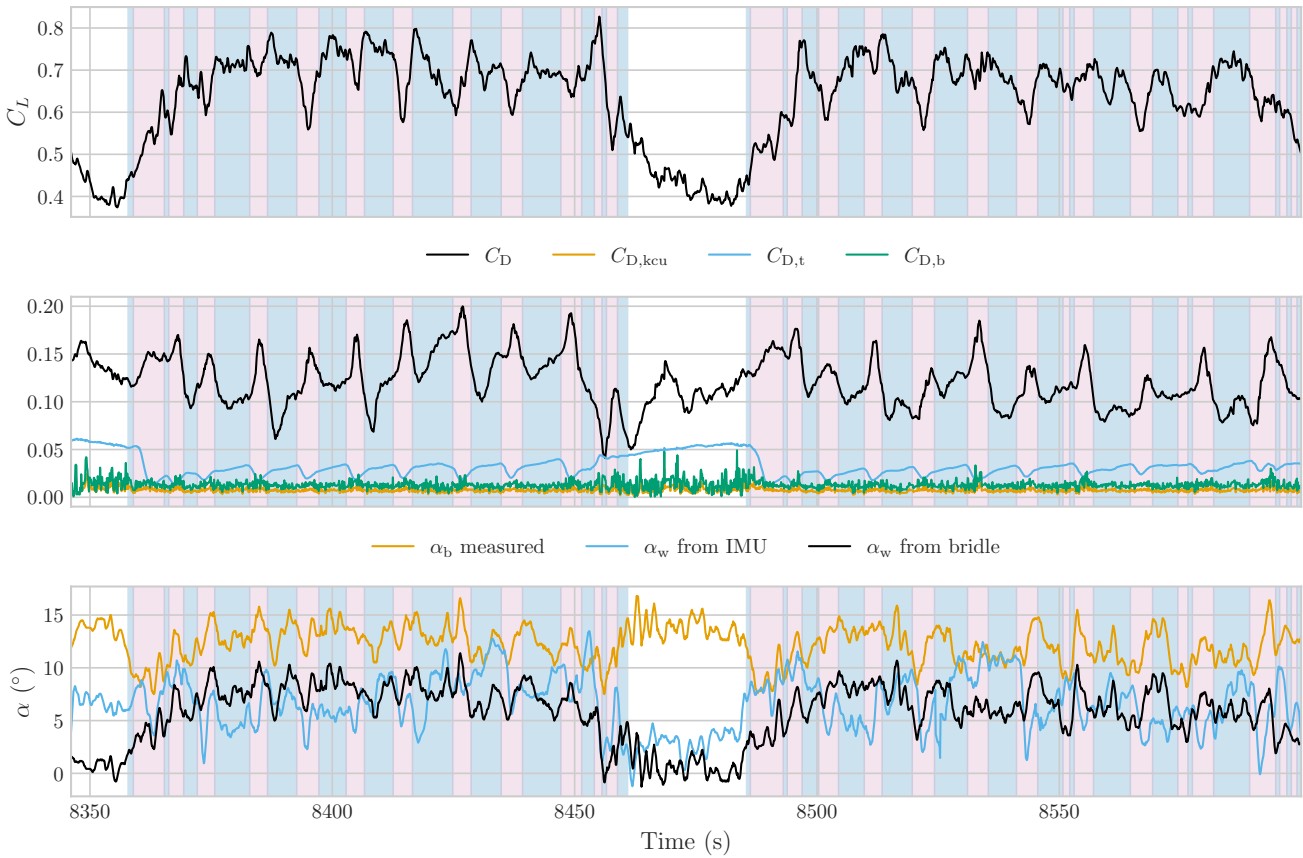

**Figure 13.** Aerodynamic coefficients and angle of attack of the V3 kite during two power generation cycles. The background colours indicate the flight phase of the kite, legend can be found in Fig. 12.

using a logarithmic wind profile, and the angle of attack was estimated based on geometric relations rather than experimentally identified deformations. The Reynolds number during this flight ranged from $2.3 \times 10^6$ to $4.5 \times 10^6$, based on the apparent airspeed and a chord length of $2.6$ m.

Compared to the RANS simulations of the CAD wing shape, a similar lift slope is observed between the mean angles of attack in reel-in and reel-out states, consistent with the VSM results. However, the EKF results show a much smaller variation in lift coefficient $C_L$ with respect to angle of attack around these mean values, indicating a flatter lift curve than predicted by both RANS and VSM, particularly below the reel-in and above the reel-out mean angles of attack. This decrease in lift slope around the mean angles of attack can be attributed to several factors. First, the aerodynamic performance of the kite is significantly influenced by steering-induced deformations, which are not captured in the rigid-wing simulations and may lead to altered polars. Second, the angle of attack measurements are affected by limited sensor accuracy and by vibrations in the wind vanes, as discussed in Sect. 3, especially when the Pitot tube shadows the vane, introducing noise that smooths the

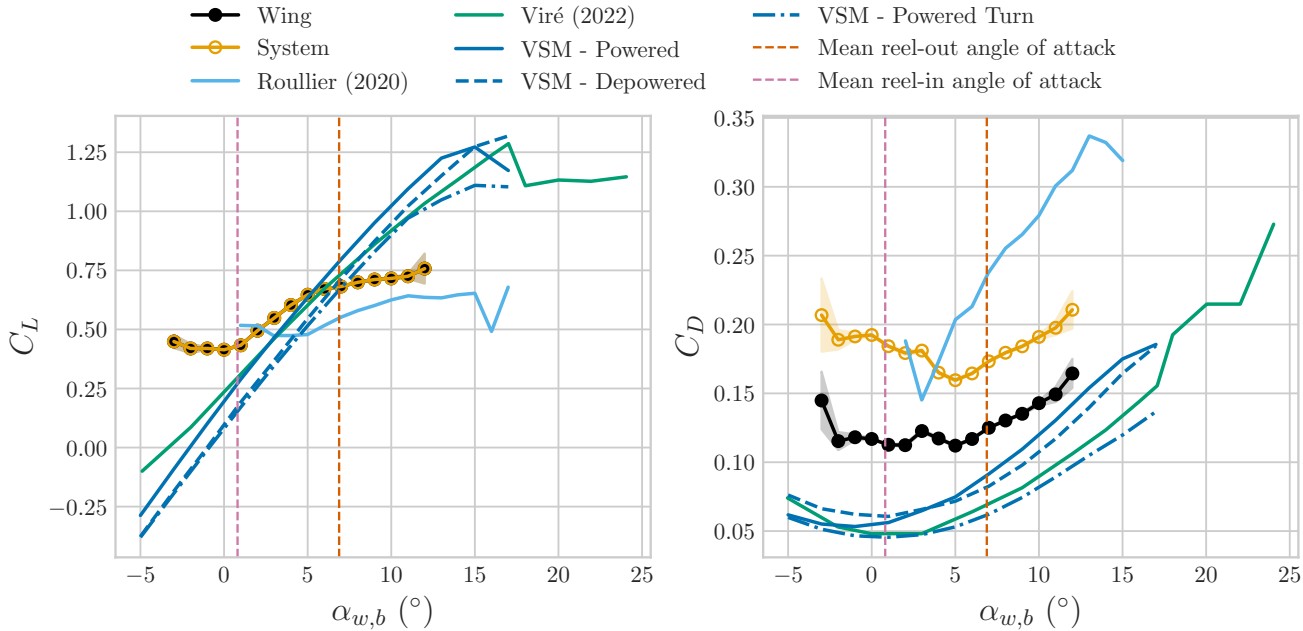

**Figure 14.** Estimated aerodynamic polars of the V3 kite from the flight on 08-10-2019 using the reconstructed wing angle of attack. The angle of attack is obtained by translating the measured angle at the bridle lines to the wing reference frame. Mean values are shown with shaded areas indicating the 99% confidence interval. Results are compared to previous studies.

curve around the trim angles. Third, CFD simulations indicate that the lift coefficient decreases with increasing sideslip angle Lebesque (2020), a phenomenon observed during flight, particularly in turning manoeuvrers.

When comparing these results to the experimental analysis by Roullier (2020), the derived polars exhibit a closer resemblance to the simulations, largely due to improvements in the estimation of both angle of attack and wind velocity. Additionally, by modelling the drag contributions from parasitic elements in more detail, the wing's aerodynamic performance can be more accurately isolated. This results in lower estimated drag coefficients and higher lift coefficients for the wing itself.

### 5.2.2 Turn dynamics

To steer the kite, a lateral force is generated by asymmetrically deforming the wing, controlled by the steering input $u_\mathrm{s}$ via actuation of the steering tape. This deformation creates a lift difference between the two sides of the wing, producing a net lateral force that steers the kite and a moment that induces yaw. Therefore, the turning behaviour can be broadly characterised by the side-force coefficient $C_S$ and the yaw rate $\dot{\psi}$.

For soft kites, where turns are predominantly induced by aerodynamic forces at the wing tips, the yaw rate can be described by a simple relationship dependent on steering input $u_\mathrm{s}$ and the apparent wind speed $v_\mathrm{a}$ (Fagiano and Novara, 2014; Erhard

and Strauch, 2012). This relationship suggests equilibrium of the aerodynamic moment during turns and is expressed as:

$$\dot{\psi} = g_k v_a \left( u_s(t - d(t)) - u_{s,0} \right), \tag{27}$$

where $g_k$ is the steering gain parameter, $u_{s,0}$ is an offset observed in the side-force coefficient estimates (Fechner, 2016), and $d(t)$ is the time delay between the steering input and the kite response (Elfert et al., 2024).

A time delay of approximately 0.1 seconds is observed when cross-correlating the yaw rate with the steering input, while the delay for the side-force coefficient is around 0.8 seconds relative to $u_s$. Understanding these delays is crucial for improving the kite responsiveness and steering precision. Further investigation is needed to determine whether these delays originate from the filter dynamics or the physical response of the kite.

In Fig. 15, the kite turn dynamics are depicted in terms of yaw rate and side force coefficient. The identified yaw rate closely matches the measured values, particularly when the offset in the steering input is accounted for. The largest discrepancies occur during straight flight sections and reel-in phases, where the kite is minimally steered. Nevertheless, as shown in Fig. 16, the yaw rate of the kite is well represented across all flight conditions by a simple turn rate law.

For the side-force coefficient, a linear relationship is fitted between the steering input and side force, providing accurate estimates during turns. However, during straight paths, a notable mismatch arises, where the side force appears to be consistently underpredicted, indicating that the same linear fit might not be suitable for all flight regimes.

This is further illustrated in Fig. 17, which shows the side force as a function of $u_s$. The data highlight distinct behaviours during turns and straight flight, with changes in side force due to the steering being more pronounced during straight flight. This phenomenon can be attributed to aerodynamic damping caused by the wing's yaw motion, which results in the outer side of the kite moving at a higher velocity relative to the inner side. Consequently, the force generated by the turn opposes the manoeuvre, altering the rate of change of the side-force coefficient $C_S$ with respect to the steering input $u_s$.

Beyond the direct effects of steering input on turn dynamics, it is also essential to consider how other components, such as the KCU, influence the overall system behaviour. By modelling the kite and KCU separately, it is possible to assess the effects of KCU inertia on the manoeuvrers and performance of the kite system. In Fig. 18, the estimated angles between the tether and the kite, defined as the pitch and roll differences between the final tether segment and the bridle segment, are shown for two power generation cycles. It is important to note that in the flight shown, the KCU was significantly oversized compared to the kite, with its weight reaching twice that of the wing itself. As a result, the behaviour observed is exaggerated compared to what would be expected in an optimised system. Nevertheless, this exaggerated scenario provides clearer insights into the effects of the KCU on the system.

During reel-out, the pitch angle remains relatively low, while the roll angle oscillates between positive and negative values, depending on the direction of the kite. During turns, where the accelerations are highest, a peak is observed in both the roll and pitch angles caused by the centrifugal force acting on the KCU. On the straight path segments of the figure eight manoeuvres, the roll angle has a lower value, primarily compensating for the weight of the KCU. However, when the kite is reeled in, due to the orientation change toward the ground station, the weight of the KCU is compensated mainly by the pitch angle.

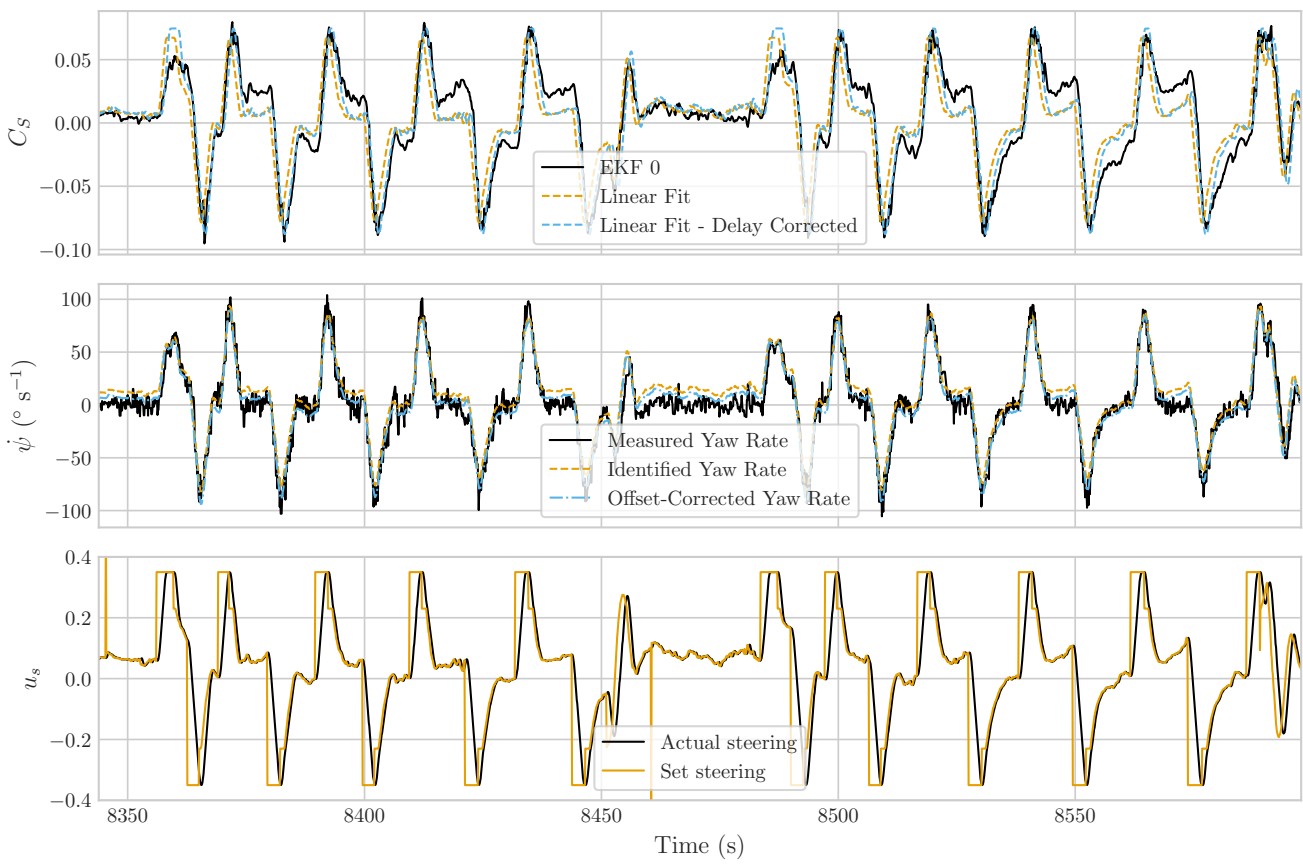

**Figure 15.** Side force coefficient, yaw rate, and steering input for two power generation cycles.

This misalignment of the kite with respect to the tether means that a portion of the aerodynamic force generated by the kite is not transmitted as tether tension but is instead used to compensate for the inertial forces at the KCU. In this exaggerated scenario, these losses can reach up to 6% during turns, highlighting the significant impact of the KCU mass on the system performance (Roullier, 2020).

### 5.3 Wind estimations

This section presents results from two selected flights from the recent flight campaign in Ireland. The 2023 flight exhibits a typical logarithmic wind profile, while the 2024 flight displays a transient phenomenon characterised by a sudden wind gust and a rapid change in wind direction. Estimates relying on a logarithmic law were obtained assuming a surface roughness length $z_0$ of 0.1 m.

For the 2023 flight, Fig. 19 shows a comparison between the wind speed and direction profiles and the results of several EKF model configurations with different sensor measurements as inputs. Ground wind measurements from a cup anemometer

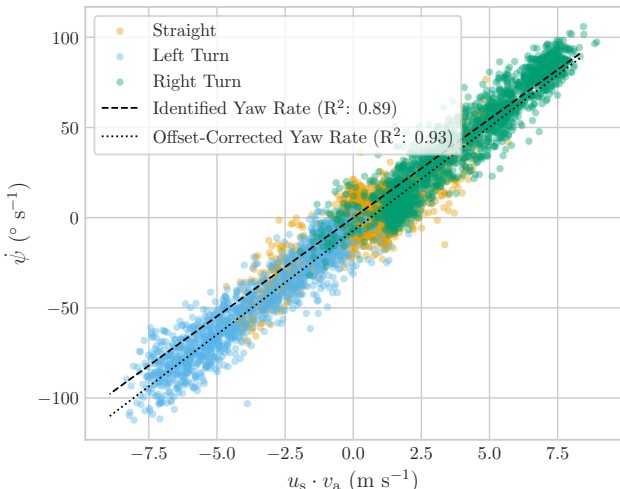

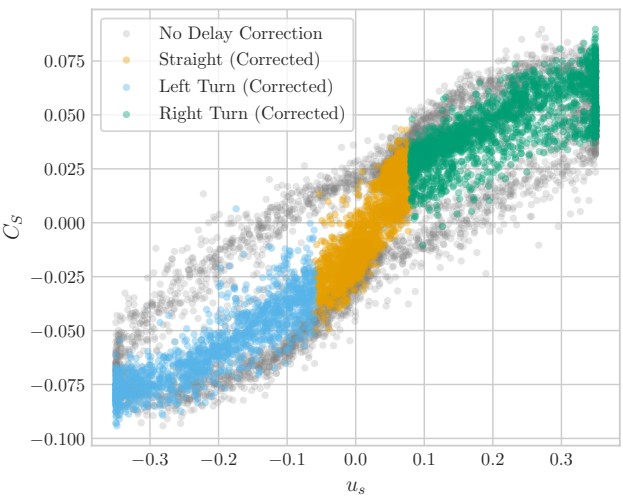

**Figure 16.** Measured yaw rate and identified turn rates with and without offset correction.

**Figure 17.** Sideforce coefficient as estimated by the EKF with and without correcting the time delay.

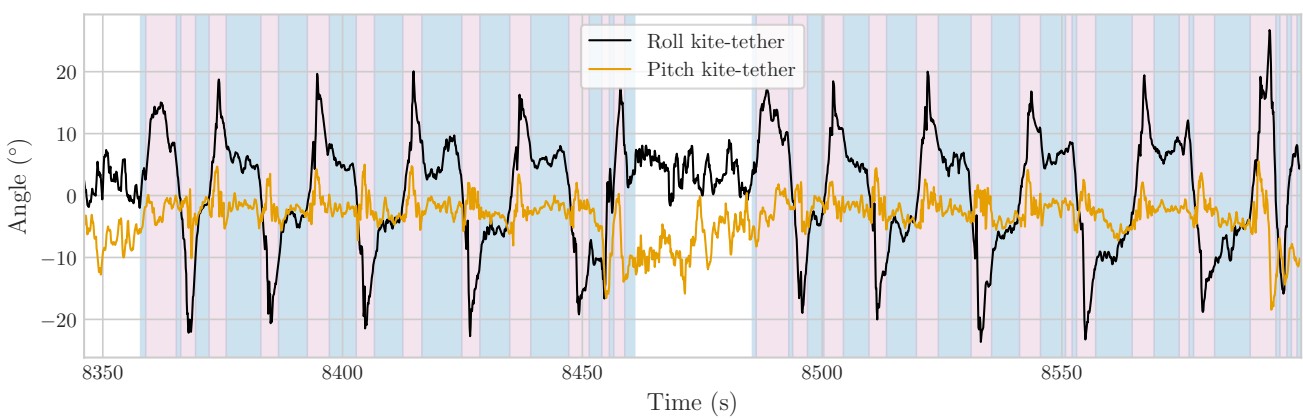

**Figure 18.** Estimated angle between tether and kite for the V3 kite during two power generation cycles. The background colors indicate the flight phase of the kite, legend can be found in Fig. 12.

and wind vanes were also available for that flight, with the wind speed extrapolated to the kite height using a logarithmic wind profile (see Equation 6). Likewise, lidar data collected at different fixed heights was interpolated to the kite altitude, with shaded areas indicating the range between minimum and maximum values. The lidar data consist of one-minute averages, with the minimum and maximum values showing variability within each averaging period.

The results, detailed in Table 4, show good agreement with the lidar for all EKF configurations for both wind speed and direction estimates. For this flight, incorporating the tether length as a measurement (EKF 1) does not lead to significant

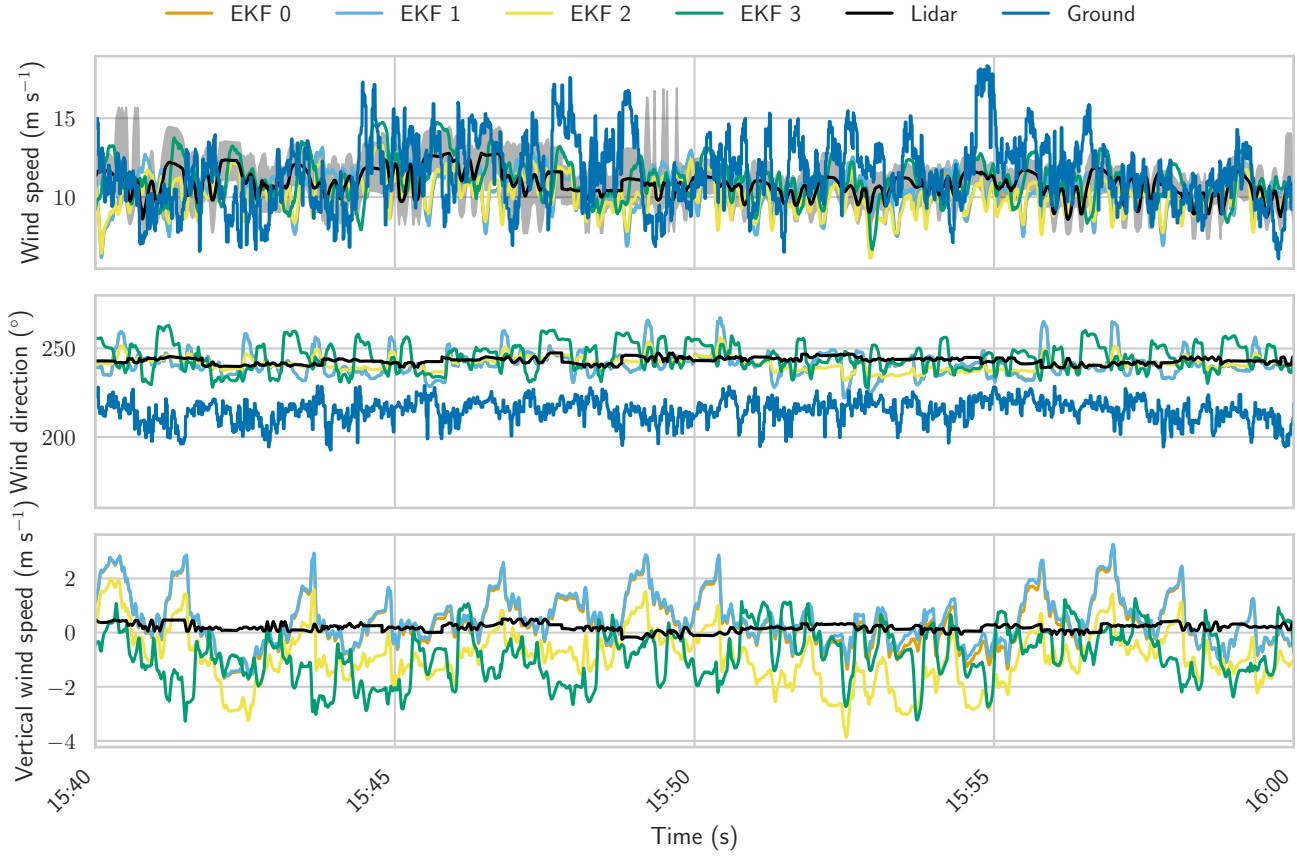

**Figure 19.** Test flight from 27-11-2023. Time series comparison of different EKF model results and lidar observations and ground measurements.

changes in the wind estimates. By modelling wind speed as logarithmically dependent on height (EKF 2), it is possible to tune wind speed and direction separately, allowing for independent control over their fluctuations. In this case, the approach resulted in slightly more accurate wind direction estimates compared to the other configurations.

The highest fluctuations in wind direction are observed when the apparent wind speed is included as a measurement (EKF 3). However, it is challenging to determine whether these fluctuations are physical or the result of errors in the Pitot tube readings, especially when compared against the lidar's one-minute averaged data. A poorly calibrated tube or its position away from the centre of rotation of the kite might cause these fluctuations. For this flight, an estimated bias of approximately $2 \text{ ms}^{-1}$ in the Pitot tube readings was identified and corrected using the methodology described in Sect. 4.2.

Despite the wind direction fluctuations, including apparent wind speed (EKF 3) yields the most accurate wind speed estimates, maintaining their accuracy even during the reel-in phase. In contrast, other configurations show increased RMSE and mean bias in this phase, resulting in a consistent underestimation of wind speed. This degradation has been attributed to two

main factors: first, a decrease in GPS data quality, as discussed in Sect. 5.1; and second, a miscalibration of the tether force measurement system, where a small offset in the elevation angle was reported. This offset has a greater impact at high elevation angles, which are more common during reel-in.

**Table 4.** The root mean squared error (RMSE) and mean bias in estimated wind direction $\phi_\mathrm{w}$ and wind speed $v_\mathrm{w}$ during reel-out and reel-in phases for four of the EKF configurations, compared to lidar measurements. Lidar data were recorded at fixed heights and matched to flight data using altitude bins of $\pm 10m$. The reel-out and reel-in phases are representative of the typical flight altitudes during these periods. Results are based on flight data from 27-11-2023. Negative biases indicate underestimation relative to lidar data. The EKF configurations with the best agreement in terms of direction and wind speed for both the reel-out and reel-in phases are highlighted in bold.

| Config. | EKF 0 | | EKF 1 | | EKF 2 | | EKF 3 | |
|---|---|---|---|---|---|---|---|---|
| RMSE \| Bias | $\phi_\mathrm{w}(°)$ | $v_\mathrm{w}(\mathrm{ms}^{-1})$ | $\phi_\mathrm{w}(°)$ | $v_\mathrm{w}(\mathrm{ms}^{-1})$ | $\phi_\mathrm{w}(°)$ | $v_\mathrm{w}(\mathrm{ms}^{-1})$ | $\phi_\mathrm{w}(°)$ | $v_\mathrm{w}(\mathrm{ms}^{-1})$ |
| **Reel-out** | 2.18 \| -1.82 | 0.76 \| 0.12 | 2.21 \| -1.84 | 0.75 \| 0.16 | **1.86 \| -1.54** | 0.71 \| -0.44 | 1.89 \| -1.58 | **0.5 \| 0.12** |
| **Reel-in** | 4.86 \| -3.77 | 1.99 \| -1.95 | 5.06 \| -3.99 | 2.03 \| -1.99 | **2.98 \| -2.58** | 1.58 \| -1.53 | 3.41 \| -3.03 | **0.29 \| 0.15** |

Regarding vertical wind speed, lidar measurements indicate an average speed close to zero, which aligns with the EKF estimates. Among the configurations, EKF 3 shows a slightly lower average vertical wind speed.

Finally, a comparison with ground-level wind measurements reveals a substantial wind veer, which the EKF effectively captures. Although the extrapolation of wind speed to the kite height provides velocities of a similar magnitude, these ground measurements cannot accurately capture wind speed variations at altitude.

The wind speed profile in Fig. 20 further confirms that incorporating apparent wind speed (EKF 3) provides the most accurate estimates. At the same time, all EKF configurations yield similarly accurate wind direction estimates, closely following the lidar measurements. The largest deviations from the lidar data occur at altitudes above 200 m, corresponding to the reel-in phase, as previously discussed in relation to Table 4.

In contrast to the typical wind profile of the 2023 flight, the 2024 flight, shown in Fig. 21, featured a wind gust starting around 13:40, accompanied by a shift in wind direction that remained fairly constant until the gust subsided. During this flight, lidar data were available at a higher resolution of one second, with measurements presented at three different heights. Additionally, measurements of the tether angles at the winch outlet were included in the model EKF 5. It is important to note that the IMU readings from the Pixhawk during this flight struggled to keep up with the high accelerations experienced during turns, resulting in clamped acceleration values and degraded measurements of the kite kinematics. This problem affected the EKF performance and led to non-physical peaks in the estimated wind velocity. Although the addition of extra measurements helped to mitigate some of these peaks—such as the one observed around 13:47—most configurations did not show significant improvements in overall accuracy.

These limitations are reflected in the estimation results, presented in Table 5. Incorporating a constraint of zero vertical wind speed (EKF 4), however, improved the wind speed estimates, which were otherwise consistently underestimated, while also reducing the overestimation of vertical wind speed relative to the lidar measurements. This effect is particularly evident in

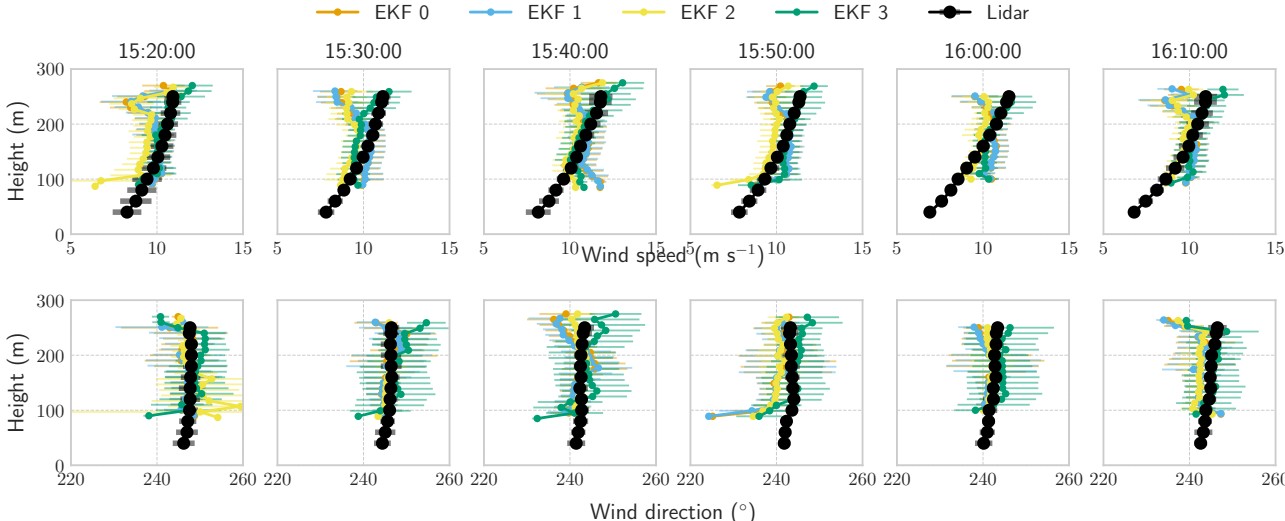

**Figure 20.** Test flight from 27-11-2023. Wind profile comparison between EKF model wind speed and direction estimates with a minimum sensor input setup, three variants with additional inputs, and lidar observations.

Fig. 21 around 13:45, where a wind gust is partially misinterpreted as an increase in vertical wind speed by all configurations except EKF 4.

Finally, and similarly to the 2023 flight, the estimates tend to degrade during the reel-in phase. This can be attributed to the same factors discussed previously, including degraded GPS quality and issues with the calibration of the tether force measurement system.

**Table 5.** The root mean squared error (RMSE) and mean bias in estimated wind direction $\phi_\mathrm{w}$ and wind speed $v_\mathrm{w}$ during reel-out and reel-in phases for four EKF configurations, compared to lidar measurements. Lidar data were recorded at fixed heights and matched to flight data using altitude bins of $\pm 10m$. The reel-out and reel-in phases are representative of the typical flight altitudes during these periods. Results are based on flight data from 05-06-2024. Negative biases indicate underestimation relative to lidar data. The EKF configurations with the best agreement in terms of direction and wind speed for both the reel-out and reel-in phases are highlighted in bold.

| Config. | EKF 0 | | EKF 2 | | EKF 4 | | EKF 5 | |
|---|---|---|---|---|---|---|---|---|
| RMSE \| Bias | $\phi_\mathrm{w}(°)$ | $v_\mathrm{w}(\mathrm{ms}^{-1})$ | $\phi_\mathrm{w}(°)$ | $v_\mathrm{w}(\mathrm{ms}^{-1})$ | $\phi_\mathrm{w}(°)$ | $v_\mathrm{w}(\mathrm{ms}^{-1})$ | $\phi_\mathrm{w}(°)$ | $v_\mathrm{w}(\mathrm{ms}^{-1})$ |
| **Reel-out** | **3.01 \| -2.18** | 1.54 \| -1.38 | 3.18 \| -2.2 | 1.70 \| -1.53 | 3.50 \| -2.58 | **0.60 \| 0.02** | 3.64 \| -2.77 | 1.43 \| -1.22 |
| **Reel-in** | 6.06 \| -5.53 | 2.19 \| -2.05 | **4.80 \| -3.88** | 2.06 \| -1.90 | 5.99 \| -5.46 | **1.53 \| -1.32** | 6.18 \| -5,54 | 1.96 \| -1.84 |

A more detailed analysis of the shape of the wind profile during the wind gust event is shown in Fig. 22. By incorporating a zero vertical wind speed measurement (EKF 4), the time response of the wind estimates improves. In contrast, other configurations show a slower time response for horizontal wind speed and instead exhibit an increase in vertical wind speed

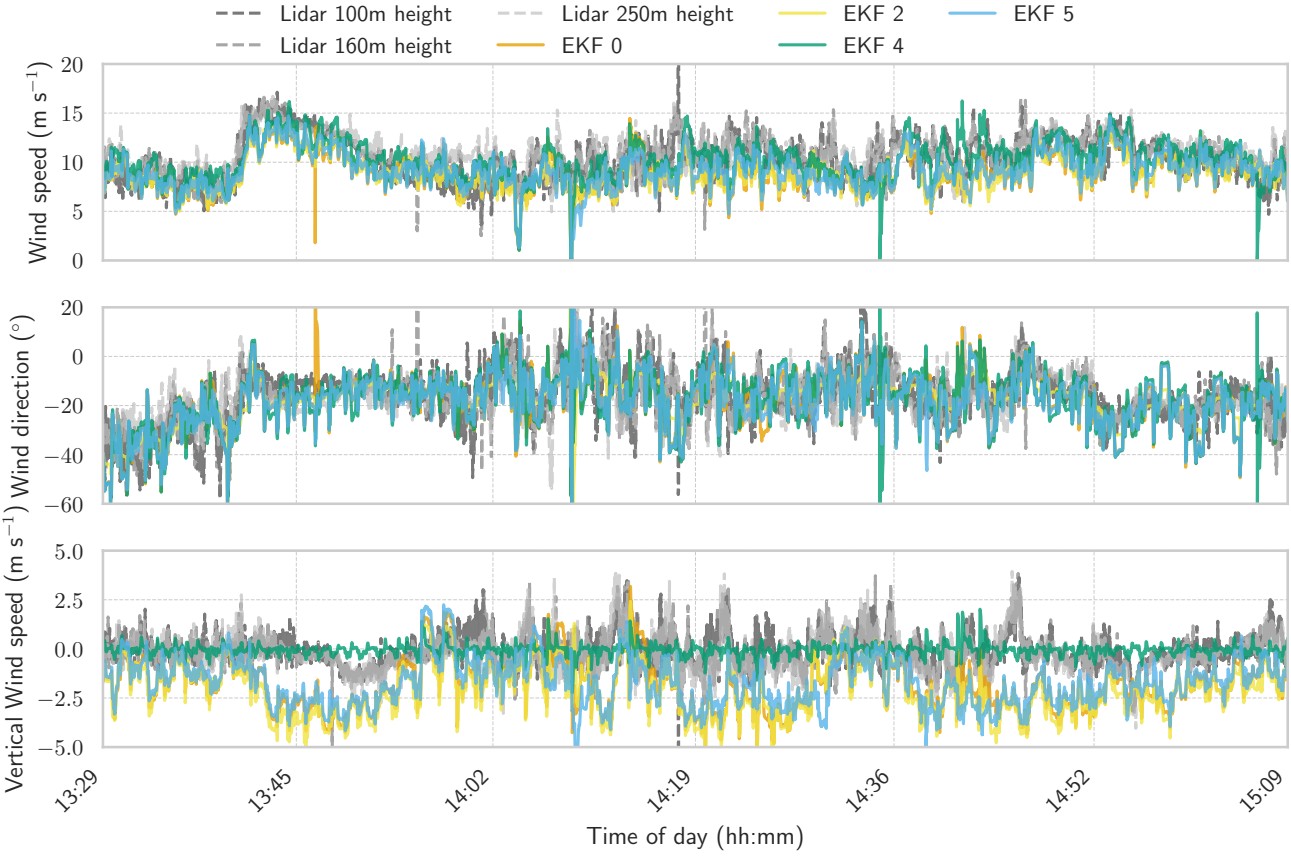

**Figure 21.** Test flight from 05-06-2024. Wind profile comparison between EKF results with a minimum sensor setup and lidar observations.

magnitude to match the same apparent wind speed. Furthermore, around 13:55, there is a localised increase in wind speed at higher altitudes that none of the models can effectively capture. Apparent wind speed measurements might have been more effective at capturing these transient phenomena, but unfortunately, they were unavailable for this flight.

Regarding wind direction, all configurations demonstrate a good time response and are able to track changes throughout the flight. Among them, the configuration using a logarithmic wind profile (EKF 2) once again provides the most accurate
estimates. The higher temporal resolution of the lidar data also allows for a more detailed comparison, confirming that the observed fluctuations in wind velocity are well captured by the models.

Overall, the 2024 flight highlights the importance of sensor reliability and the potential of constrained models like EKF 4 to improve robustness under dynamic wind conditions.

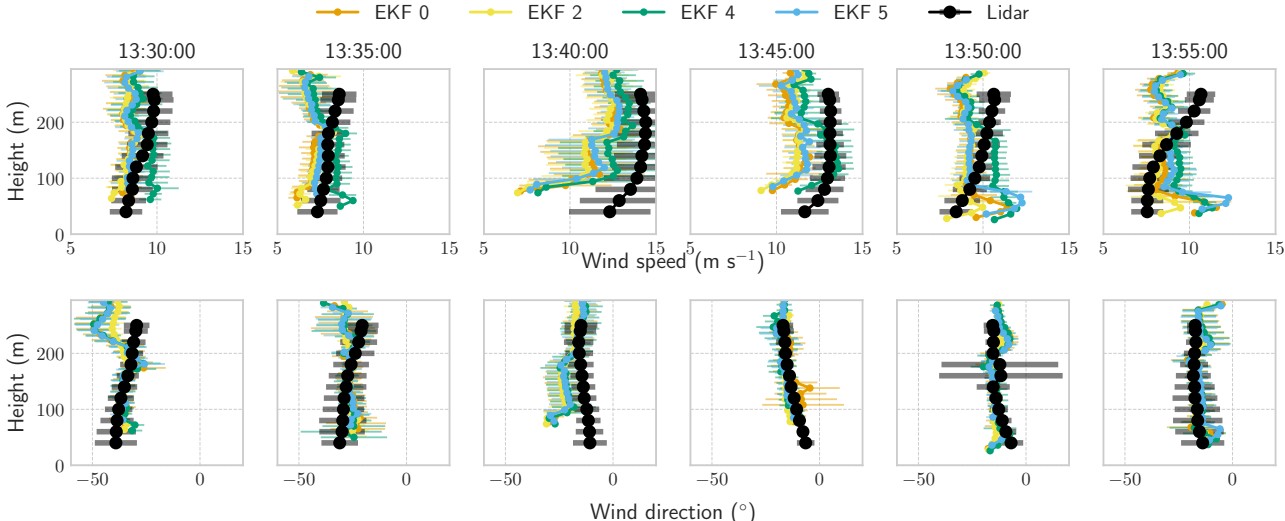

**Figure 22.** Test flight from 05-06-2024. Wind profile comparison between EKF results with a minimum sensor setup and lidar observations.

## 5.4 Turbulence measurements

Since the EKF estimates wind characteristics based on measurements that are not directly related to the inflow, accurate estimates of the variability in wind speed are highly dependent on the correct tuning of the filter. To verify that the rapid changes estimated are physical, one can examine the turbulence estimates, which quantify these variations.

A key aspect to investigate is the power density spectrum of the wind speed, as shown in Fig. 23. The sampling frequency of the measurements was $0.1\,\mathrm{s}$, corresponding to frequencies resolved of up to $5\,\mathrm{Hz}$. It is observed that the energy cascade

follows the Kolmogorov slope within the range of $\sim 0.01\,\mathrm{Hz} - 0.3\,\mathrm{Hz}$, with higher frequencies damped, likely due to the time response of the EKF. Additionally, a peak is observed below $0.01\,\mathrm{Hz}$, corresponding to the cycle timescale for the analysed flight. This peak coincides with a rapid change in wind speed, attributed to the significant variation in kite height experienced during reel-in.

Another parameter assessed in the study is the turbulence intensity, defined as the standard deviation of the wind speed

divided by its mean value. This parameter is commonly used in conventional wind energy assessments, particularly when assessing turbine fatigue loads. Turbulence intensity is also measured by the profiling lidar, providing an additional validation of the EKF. However, it is important to note that the quantities measured by the lidar and the EKF are not directly equivalent.

The lidar measures turbulence intensity over an averaged volume at a fixed height, with the area determined by the laser cone angle. In contrast, the turbulence intensity derived from the EKF is calculated over a range of heights within $\pm 10\mathrm{m}$ of the

690 lidar measurement to obtain comparable values. It is calculated as the 1-minute standard deviation divided by the mean.

With these considerations in mind, turbulence intensity estimates from both the EKF and lidar are compared in Fig. 24 for the two analysed flights. The comparison reveals good agreement in magnitude, and the temporal behaviour of the EKF closely

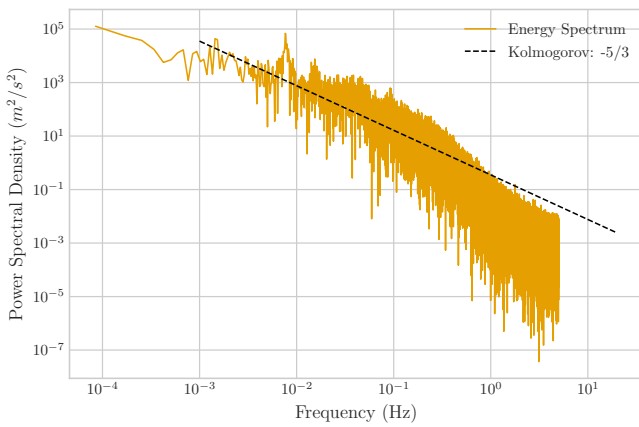

**Figure 23.** Power spectral density of the wind speed. Test flight from 08-10-2019.

matches that of the lidar, as illustrated in Fig. 24b. While these initial comparisons are promising, a more detailed analysis is needed to fully assess the accuracy of these measurements.

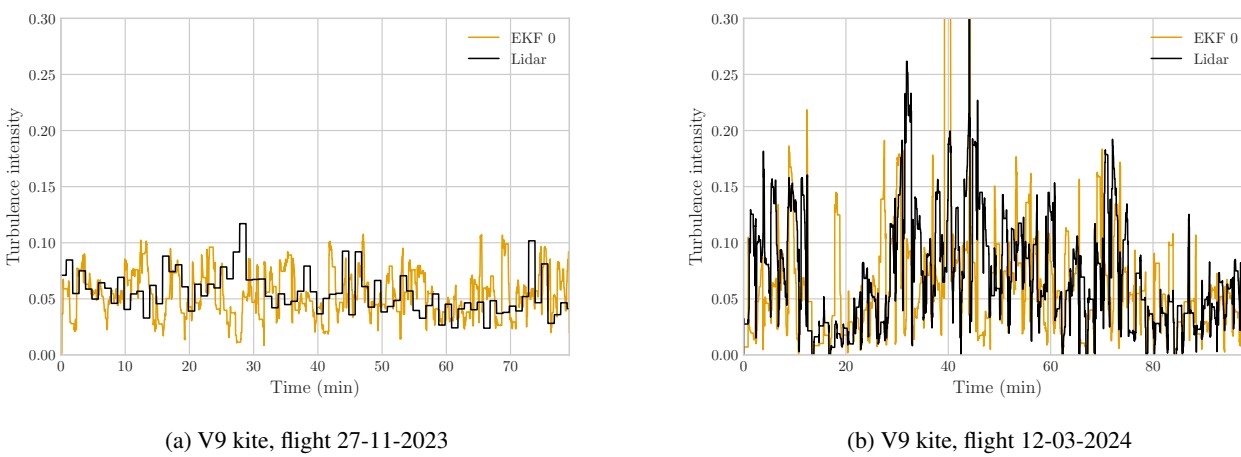

(a) V9 kite, flight 27-11-2023        (b) V9 kite, flight 12-03-2024

**Figure 24.** Turbulence intensity at a height of 140m

**6 Discussion and conclusions**

This study presents a sensor fusion technique for tethered flying systems to estimate the state of the system and the wind conditions at the kite. The system state includes the kite's position, velocity, aerodynamic performance, and tether shape. The sensor fusion technique consists of an iterated extended Kalman filter (EKF), modelling the kite as a point mass and the tether

as a system of point masses linked by spring damper elements. The tether shape is assumed to be quasi-static and includes the kite control unit (KCU) as a point mass, linked to the kite by an additional spring damper element representing the bridle line system. By integrating data from multiple sensors, such as position, velocity, tether force, and reeling speed, the EKF model can accurately estimate system dynamics and wind conditions without the need of direct airflow measurements, provided the filter is appropriately tuned to the specific system.

The proposed EKF effectively estimates average wind profiles, particularly those that follow a conventional logarithmic slope. Its performance is less accurate in scenarios with significant deviations, such as sudden wind gusts, where the filter requires a few minutes to adapt to rapid changes in wind speed. Notably, the filter adapts faster to changes in wind direction, indicating greater sensitivity to directional variations. To improve the time resolution in wind estimations, direct airflow measurements should be included, such as from a well-calibrated and maintained Pitot tube.

The accuracy of the estimations depends on the quality and reliability of the input measurements. While the EKF can compensate for minor inaccuracies and correct some biases, problems, such as improperly calibrated sensors, signal loss, or clamped acceleration signals, can degrade performance. This was observed in the 2024 flight, where acceleration measurements saturated during high-speed turns, contributing to a reduced estimation accuracy.

Sensor noise and filter tuning are areas that deserve further attention. While the paper acknowledges the potential inaccuracies stemming from incorrect tuning and noise modelling, it does not discuss strategies to mitigate these problems in-depth. For the proposed EKF to be more robust for a specific system, it can be beneficial to incorporate a more comprehensive noise modelling. Implementing advanced calibration techniques, such as adaptive filtering methods for real-time sensor noise adjustments, could enhance the accuracy and reliability of state estimations.

The proposed approach also demonstrates the ability to estimate tether and kite orientations, which closely align with the measurements at the kite. This capability enables the estimation of pitch changes induced by deformation around the central struts of the wing, typically referred to as the depower angle. This information can be used to estimate the wing angle of attack based on angle of attack measurements in the bridle line system. The kite was aerodynamically characterized for the range of flight conditions available using this newly estimated angle, showing an improvement with respect to previous analyses. However, for a complete aerodynamic characterization, tailored test flights must be conducted to capture the aerodynamic behaviour of the kite across different conditions.

This study explored several sensor setups, demonstrating that a single airborne sensor measuring the kite position and velocity, combined with ground-based force and tether length sensors, is sufficient to obtain reasonable estimates, as supported by validation against lidar data. While modeling the wind profile as logarithmic does not necessarily improve the estimates, it allows for independent tuning of the wind direction and the magnitude of the process noise, which can be useful for supervisory control applications that require damping of high-frequency variations. Furthermore, including a soft constraint on vertical wind speed—implemented as a pseudo-measurement centred at zero—was shown to improve the filter's time response during transient events (e.g. wind gusts), by reducing the misinterpretation of horizontal wind changes as vertical components.

For applications requiring high time-resolution wind estimation, installing well-calibrated flow sensors on the kite is recommended. In the absence of direct inflow measurements, the wind must be inferred indirectly from forces and dynamics,

resulting in lower time resolution. Additionally, the EKF implementation was benchmarked on a standard laptop and shown to run over 50 times faster than real time with modest CPU and memory usage (see Appendix C1). While further optimisation would be required for embedded onboard deployment, these results indicate that real-time implementation is feasible.

Overall, the EKF has proven to be a robust method for wind and state estimation for AWESs, demonstrating sufficient accuracy for mean wind speed and direction estimation for power generation cycles. Across the evaluated flights, average RMSE values for wind speed estimation ranged between 0.3 and 2.2 $\mathrm{ms^{-1}}$, with biases below 0.2 $\mathrm{ms^{-1}}$ in the best performing configurations. Directional RMSEs remained typically below 5°, including during transient wind events. However, its responsiveness to rapid changes in wind conditions is limited by the quality and availability of sensor data. Future work should focus on optimising sensor setups and conducting targeted system identification test flights to capture dynamic states beyond normal operational limits, ensuring the kite can be characterized fully. Ultimately, this study highlights that the kite can be effectively used as a sensor, providing valuable insights into system dynamics and wind conditions.

*Code and data availability.* The code can be found in the attached Github repository. It is currently private, but will be made public when the paper is published. https://github.com/ocayon/EKF-AWE.

The datasets can be found on different data repositories: 1. Flight data 08-10-2019, Schelbergen et al. (2024). 2. Flight data 27-11-2023, Cayon et al. (2024a). 3. Flight data 05-06-2024, Cayon et al. (2024b). The flights from 2023 and 2024 are under one year embargo and will be public in November 2025.

## Appendix A: Appendix A: Nomenclature

Nomenclature used throughout this study. Bold symbols denote vectors.

### Kite and Tether Model Variables

**r** Position in the ENU frame (m)

**v** Velocity in the ENU frame ($\mathrm{ms^{-1}}$)

**a** Acceleration in the ENU frame ($\mathrm{ms^{-2}}$)

$\mathbf{F}_{t,k}$ Tether force at the kite (N)

$\mathbf{F}_{t,g}$ Tether force at the ground station (N)

$\mathbf{F}_{a,k}$ Aerodynamic force on the kite (N)

$\mathbf{F}_{g,k}$ Gravitational force on the kite (N)

$\mathbf{v}_a$ Apparent wind velocity ($\mathrm{ms^{-1}}$)

$\mathbf{v}_w$  Ambient wind velocity $(\mathrm{ms}^{-1})$

$l_t$  Tether length (m)

$\beta_0$  Elevation angle of first tether segment (°)

$\phi_0$  Azimuth angle of first tether segment (°)

  $\phi_k$  Roll Euler angle in NED frame (°)

$\theta_k$  Pitch Euler angle in NED frame (°)

$\psi_k$  Yaw Euler angle in NED frame (°)

$\dot{\psi}$  Yaw angle rate $(°\mathrm{s}^{-1})$

## Aerodynamic Parameters

  $C_L$  Lift coefficient

$C_D$  Drag coefficient

$C_S$  Side-force coefficient

$A_k$  Projected wing area $(\mathrm{m}^2)$

$\rho$  Air density $(\mathrm{kgm}^{-3})$

  $\alpha_b$  Angle of attack at the bridle (°)

$\alpha_w$  Angle of attack at the wing (°)

$\alpha_d$  Depower angle (°)

$u_p$  Depower input (0–1)

$\lambda_0$  Angle between tether and power lines (°)

  **Tether Model Parameters**

$\mathbf{f}_{D,j}$  Aerodynamic drag on tether segment $j$ (N)

$\mathbf{f}_{L,j}$  Aerodynamic lift on tether segment $j$ (N)

$d_t$  Tether diameter (m)

$C_\perp$  Crosswise drag coefficient of the tether

$C_\parallel$ Axial drag coefficient of the tether

$\alpha_j$ Local angle of attack of tether segment $j$ (°)

$C_{\perp,\mathrm{kcu}}, C_{\parallel,\mathrm{kcu}}$ Crosswise and axial drag coefficients of the KCU [–]

**Filter and Control Variables**

$\mathbf{x}$ State vector of the EKF

$\mathbf{u}$ Input vector of the EKF

$\mathbf{z}$ Measurement vector of the EKF

$\mathbf{P}$ State estimation error covariance matrix

$\mathbf{Q}$ Process noise covariance matrix

$\mathbf{R}$ Measurement noise covariance matrix

$\mathbf{K}$ Kalman gain matrix

$\delta$ Sensor bias

$\eta$ Measurement noise

$u_\mathrm{s}$ Steering input

$g_\mathrm{k}$ Steering gain

$d(t)$ Time delay between steering input and response (s)

$u_{\mathrm{s},0}$ Steering input offset [–]

**Other**

$z$ Height above ground (m)

$z_0$ Surface roughness length (m)

$u^*$ Friction velocity (ms$^{-1}$)

$\phi_\mathrm{w}$ Wind direction (°)

$v_\mathrm{w}^z$ Vertical component of wind velocity (m/s)

TI Turbulence intensity ($\sigma_v/\bar{v}$) [–]

$\kappa$ von Kármán constant ($\approx 0.4$)

## Appendix B: Appendix B: System specifications and tuning parameters

This appendix provides details on the AWES configuration and the EKF tuning parameters used in this study.

### B1 System description

This section provides the key physical specifications of the kite, the KCU, and the tether used in the airborne wind energy system.

**Table B1.** Kite specifications (model V3).

| Parameter | Value | Unit |
|---|---|---|
| Model name | V3 | – |
| Mass | 15 | kg |
| Area | 19.75 | m$^2$ |
| Span | 10 | m |

**Table B2.** Bridle and Kite Control Unit (KCU) parameters.

| Parameter | Value | Unit |
|---|---|---|
| KCU length | 1 | m |
| KCU diameter | 0.48 | m |
| $C_{\perp,\mathrm{kcu}}$ | 0.69 | – |
| $C_{\parallel,\mathrm{kcu}}$ | 0.83 | – |
| Mass | 27.6 | kg |
| Distance to kite | 11.5 | m |
| Total bridle length | 96 | m |
| Bridle line diameter | 0.0025 | m |

### B2 Tuning parameters for EKF

This section lists the standard deviations used in the EKF, both for sensor measurements and model process noise.

## Appendix C: Appendix C: Kalman filter performance

This section presents performance metrics of the implemented Kalman filter and details the computational environment used for testing and validation.

**Table B3.** Tether parameters.

| Parameter | Value | Unit |
|---|---|---|
| Material | Dyneema-SK78 | – |
| Density ($\rho_t$) | 970 | kg m$^{-3}$ |
| Young's modulus (E) | 132 | GPa |
| Diameter | 0.01 | m |
| $C_\perp$ | 1.1 | – |
| $C_\parallel$ | 0.01 | – |
| Number of elements | 10 | – |

**Table B4.** Measurement standard deviations used in the EKF.

| Parameter | Value | Unit |
|---|---|---|
| Position ($\mathbf{r}_k$) | 5 | m |
| Velocity ($\mathbf{v}_k$) | 2 | m/s |
| Tether length ($l$) | 0.5 | m |
| Tether elevation ($\beta_0$) | 3 | ° |
| Tether azimuth ($\phi_0$) | 3 | ° |
| Position (tether model) | $1 \times 10^{-5}$ | m |
| Zero vertical wind speed ($v_\mathrm{w}^\mathrm{z} = 0$) | 2 | m/s |
| Apparent airspeed ($v_\mathrm{a}$) | 1 | m/s |
| Bridle angle of attack ($\alpha_b$) | 4 | ° |

*Author contributions.* Conceptualisation, O.C. and R.S.; methodology, O.C.; software, O.C.; investigation, O.C.; writing—original draft preparation, O.C.; writing—review and editing, O.C., R.S. and S.W.; supervision, R.S. and S.W.; funding acquisition, R.S. and S.W. All authors have read and agreed to the published version of the manuscript.

*Competing interests.* At least one of the (co-)authors is a member of the editorial board of Wind Energy Science. R.S. is a co-founder of and advisor for the start-up company Kitepower B.V., which is commercially developing a 100 kW kite power system and provided their test facilities and staff for performing the in situ measurements described in this article. Both authors were financially supported by the European Union's Meridional project, which also provided funding for Kitepower B.V.

**Table B5.** Model standard deviations used in the EKF.

| Parameter | Value | Unit |
|---|---|---|
| Position ($\mathbf{r}_k$) | 2.5 | m |
| Velocity ($\mathbf{v}_k$) | 1 | m/s |
| Wind velocity ($\mathbf{v}_w$) | 0.1 | m/s |
| Friction velocity ($u_*$) | 0.002 | m/s |
| Wind direction ($\phi_\mathrm{w}$) | 0.2 | ° |
| Vertical wind speed ($v_\mathrm{w}^\mathrm{z}$) | 0.01 | m/s |
| Lift coefficient ($C_L$) | 0.01 | – |
| Drag coefficient ($C_D$) | 0.003 | – |
| Side-force coefficient ($C_S$) | 0.01 | – |
| Tether elevation ($\beta_0$) | 5 | ° |
| Tether azimuth ($\phi_0$) | 5 | ° |
| Tether length ($l$) | 0.1 | m |

**Table C1.** Kalman filter performance metrics.

| Metric | Value |
|---|---|
| Average iteration time | 0.0025 s |
| CPU usage | 13% |
| Real-time performance | 50× faster than real-time |
| Peak memory usage | 169.54 MB |

**Table C2.** Computational environment specifications.

| Component | Specification |
|---|---|
| CPU | 12th Gen Intel Core i7-1265U (10 cores, 12 threads) |
| RAM | 16 GB |
| Operating system | Windows 10 64-bit |
| Software environment | Python 3.10, NumPy 1.24, CasADi 3.6.0 |

*Acknowledgements.* This work has been supported by the MERIDIONAL project, which receives funding from the European Union's Horizon Europe Programme under the grant agreement No. 101084216. We acknowledge the use of OpenAI's ChatGPT and Grammarly for assistance in refining the writing style of this manuscript.

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
