# Peer review of "Kite as a Sensor: Wind and State Estimation in Tethered Flying Systems"

_Wind Energy Science, 2024_

## Referee Comment (RC1)

Further development and optimization of Airborne Wind Energy Systems requires of advance characterization and monitorization of tethered aircrafts flight dynamics. However, this can result challenging specially for flexible soft kites, as they are highly susceptible to changes in wind speed and direction. The paper presents a sensor fusion technique which is an evolution of previous works based on EKFs, effectively estimating wind velocity at flying altitude using the kite as a sensor. Albeit there is no significant novelty in the theory of the methods used, the application of the EKF and the collected experimental data, in particular, the implementation of a model that accounts for both tether sag and kite control unit inertia and aerodynamics, should be interesting for the community.

Overall, the article is well written, figures are high quality and the mathematical formulations and the technical information are presented accordingly. However, I have one major concern for the publication of this paper. For me is very difficult to follow the reasoning about the EKF implementation/design, specially the sensors used and the observation model of both the V3 and V9 kites flight tests. For example:

> 175-*"Overall, the sensors that are least susceptible to the intrinsic deformations of the soft kite and the high accelerations of the system, and thus more reliable, are the GPS, the load cell (for tether force), and the mechanism measuring the tether length and tether angles. These sensors can maintain their accuracy despite the flexible nature of the kite. Consequently, the proposed sensor fusion model primarily relies on these measurements, resulting in a* **minimal sensor setup consisting of a GPS (for position and velocity), a load cell, and a tether length measurement**".

> 311-**"The required minimum measurements are the position and velocity of the kite wing"**

> 393- *"In this section, we explore various sensor setups and model configurations. The different EKF models are detailed in Table 2. The additional measurements listed in the table are used alongside the* **minimum required sensors for a system with a KCU, which include the position, velocity, acceleration of the kite wing, tether force, and reel-out speed.**"

In my opinion, the paper should be optimized for increased clarity and conciseness. The authors should make an effort to facilitate the reader the matching between caps 4 and 5.

I also have some minor comments:

- Line 115-120, further discussion about direct measurement of in-situ aerodynamic angles of attack and sideslip is welcomed (Oehler and Schmehl, 2019). Using booms for isolating aerodynamic sensors from aircraft's perturbations is a well-known practice in the aerospace industry during development phases.
- Line 143 → Fig2. could be improved by showing a detailed view/scheme of the implementation of the load cell sensing the tension of the tether without interfering with the reel in-out system.
- Further detail about how airborne data is logged/transmitted to the ground and synchronized with in the ground measured data is also welcomed.

- Line 396, 405-410 → In my opinion, the output of the Px4 position and velocity estimations should not be used as measurements for the EKF as errors are not guaranteed to be zero mean and gaussian. Instead, raw GPS position and velocity from PixHawk Gps should be used, plus a measurement error model to guarantee that the measurement noise described in eq.24a and 24b is Gaussian white noise. (R.Borobia et al. 2018). This change will eliminate the dynamics of the PixHawk onboard estimator increasing the stability of the filter.
- Line 411 → In Fig9. The measured Euler angles are the estimated ones by Px4?
- Line 419 → Calculation of Yaw angle assuming alignment of the kite body axis with aerodynamic velocity vector assumes no side-slip during the flight. However, direct measurement of side-slip angle showed non zero values for a inflatable kite (R.Borobia et al. 2021)
- 520 -530 → The underpredicted side-force could be related to assuming zero-side slip angle?

---

## Referee Comment (RC2)

[referee-annotated manuscript omitted]

---

## Author Response (AR1)

**Response to Reviewer 1**

**April 4, 2025**

**Major Concern:**

*"Overall, the article is well written, figures are high quality and the mathematical formulations and the technical information are presented accordingly. However, I have one major concern for the publication of this paper. For me is very difficult to follow the reasoning about the EKF implementation/design, specially the sensors used and the observation model of both the V3 and V9 kites flight tests.*

*In my opinion, the paper should be optimized for increased clarity and conciseness. The authors should make an effort to facilitate the reader the matching between caps 4 and 5."*

**Response:** We thank the reviewer for highlighting the need for greater clarity in the explanation of the EKF design and its relationship to the sensor configurations used in the V3 and V9 kite flight tests.

To address this concern, we have revised Sections 3 to 5 to explicitly distinguish between required and optional measurements across different EKF configurations. In particular:

- In Section 3 (Sensor Setup), we clarify more concisely the most reliable measurements, which include position and velocity (from GPS), tether force (from a load cell), and reel-out speed (from the tether reel-out encoder), which are sufficient to estimate the kite motion. We also specify that when the KCU is modeled, a measurement of acceleration is additionally required to account for its inertial effects.

- In Section 4 (Filter Design), we state more clearly the required measurements for the EKF, and the need for additional acceleration measurement if the KCU is modeled.

- In Section 5, we improved the explanation of the different EKF setups and explicitly reference Section 3 when introducing the baseline configuration. This reinforces the logic behind the sensor combinations used in the evaluation of V3 and V9 datasets.

Together, these changes clarify the reasoning behind the EKF design and improve the consistency of terminology across sections. We hope these improvements resolve the reviewer's concern and make the EKF framework easier to follow.

**Minor Comment 1:**

*"Line 115-120, further discussion about direct measurement of in-situ aerodynamic angles of attack and sideslip is welcomed (Oehler and Schmehl, 2019). Using booms for isolating aerodynamic sensors from aircraft's perturbations is a well-known practice in the aerospace industry during development phases"*

**Response:** We thank the reviewer for this helpful suggestion. We have expanded the corresponding paragraph in the sensor setup section to include the resolution of the sensor. In response to the reviewer's comment, we now mention the use of boom-mounted sensors on the leading edge of the kite during development phases, which is indeed a common approach in the aerospace industry. While effective for isolating sensors from the perturbations caused by the wing, this solution is generally unsuitable for commercial airborne wind energy systems due to its fragility. The revised paragraph also includes references to Oehler and Schmehl (2019) and Borobia-Moreno et al. (2021) to support these points.

**Minor Comment 2:**

*"Line 143 → Fig2. could be improved by showing a detailed view/scheme of the implementation of the load cell sensing the tension of the tether without interfering with the reel in-out system."*

**Response:** We appreciate the reviewer's suggestion. However, we prefer not to include a detailed schematic of the load cell implementation due to its proprietary nature. A schematic of a similar system can be found in Hummel (2018), which illustrates the principle used. We trust the updated description in the manuscript clarifies the sensing approach while respecting confidentiality constraints.

**Minor Comment 3:**

*"Further detail about how airborne data is logged/transmitted to the ground and synchronized with the ground-measured data is also welcomed."*

**Response:** We appreciate the reviewer's interest in the data acquisition and synchronization procedures. However, the airborne and ground-based sensor data used in this study were pre-processed and provided by Kitepower, and the specifics of the logging, transmission, and synchronization infrastructure were not available to the authors. As such, we are unable to provide further technical detail on this aspect. We would also like to note that the focus of this work is on the development and evaluation of the EKF estimation framework, which operates on already time-aligned datasets. As a result, the underlying transmission architecture does not impact the methodology or the conclusions of this study.

**Minor Comment 4:**

*"Line 396, 405-410 → In my opinion, the output of the Px4 position and velocity estimations should not be used as measurements for the EKF as errors are not guaranteed to be zero mean and gaussian. Instead, raw GPS position and velocity from PixHawk Gps should be used, plus a measurement error model to guarantee that the measurement noise described in eq.24a and 24b is Gaussian white noise. (R.Borobia et al. 2018). This change will eliminate the dynamics of the PixHawk onboard estimator increasing the stability of the filter."*

**Response:** We fully agree with the reviewer that using raw GPS measurements would be preferable to ensure that the assumptions on the measurement noise in the observation model (Eq. 24a and 24b) hold. Unfortunately, in the dataset analysed, only the output of the Pixhawk's internal EKF (GPS+IMU fused) was logged, and the raw GPS data were not recorded.

We attempted to reflect this limitation in the manuscript (lines 396–397), but we recognise that the implication for the filter's noise characteristics and stability may not have been sufficiently emphasised. To address this, we have clarified the text accordingly to highlight the impact of using pre-filtered measurements.

**Minor Comment 5:**

*"Line 411 → In Fig9. The measured Euler angles are the estimated ones by Px4?"*

**Response:** Yes, the Euler angles shown in Fig. 9 are the estimates provided by the PX4 onboard EKF, based on the Pixhawk's IMU data. A clarification is added to the text to make this point explicit and avoid future readers to question the same.

**Minor Comment 6:**

*"Line 419 → Calculation of Yaw angle assuming alignment of the kite body axis with aerodynamic velocity vector assumes no side-slip during the flight. However, direct measurement of side-slip angle showed non zero values for a inflatable kite (R.Borobia et al. 2021)"*

**Response:** We agree with the reviewer that the assumption of zero side-slip does not strictly hold, particularly during manoeuvres. Our intention was not to neglect sideslip effects, but rather to illustrate the general tendency of the kite to align with the local inflow. To clarify this point, we have extended the discussion by including measured sideslip angle statistics from the V9 dataset, which show a standard deviation of approximately $2.5°$ and peak values up to $5°$ during turns.

**Minor Comment 7:**

*"520 -530 → The underpredicted side-force could be related to assuming zero-side slip angle? "*
**Response:**

We thank the reviewer for the observation. To clarify, the side-force is estimated by the EKF, but it is not modelled as a function of the side-slip angle (see section 4). The dynamic model employed is a 3-DOF point-mass formulation, where yaw dynamics are not considered. The kite orientation (pitch and roll) is inferred from the bridle segment orientation in the quasi-static tether model, rather than modelled as a full 6-DOF body. The assumption of alignment with the apparent wind is used only in post-processing to estimate the yaw angle for qualitative analysis and is not part of the EKF or the dynamic model. Therefore, the underprediction of the side-force is not attributable to a zero side-slip assumption.

**Response to Reviewer 2**

April 4, 2025

**Major Comments:**

*"This paper addresses some very interesting questions on kite dynamics using fusion techniques for the analysis of experiments. It draws on the substantial experience of the Delft team, which is undoubtedly the world's leading centre for kite studies. The paper is composed of 41 pages including 4 pages only for the introduction. Section 2, 3 and 4 present the material, i.e. system, sensors and fusion technics used for analysis. Section 5 presents many results in 4 subsections (kite kinematics, system dynamics (aerodynamic identification and turning dynamics), wind estimation and turbulence measurements). It ends with a broad conclusion.*

*The article could probably have been split into 2 separate articles. That would have made the point clearer. The language is highly technical, which often makes it difficult to read. It is often difficult to find the definition of one of the many variables. In such cases, an appropriate nomenclature is essential. Given the complexity of the problem being addressed and the large number of variables involved, the use of full variable names in the text should be preferred most of the time for easier reading.*

*One of the main gap in the document is a clear definition of the reference frames used in this study. With all the information available, the reader probably has all the information needed to find the definition of each variable. But this definition is uncertain, and the reader may make a mistake. An appendix at the end of the document gives the values of the experimental parameters and the model used during the tests. This should ensure reproducibility of the results. The codes are also provided."*

**Response:**

We are grateful to the reviewer for the thoughtful and encouraging comments on our manuscript. We appreciate the acknowledgement of the Delft team's longstanding contributions to the field and the recognition of the paper's ambition in consolidating these insights.

We acknowledge that the manuscript is lengthy; however, we consider the topic to be a closed and coherent body of work. The aim was to integrate a broad range of experimental and modelling expertise into a single, comprehensive study. Given the depth of analysis and the novel fusion framework presented, we believe this integrated presentation will serve the airborne wind energy community more effectively than a fragmented approach across multiple papers.

In response to the specific points raised:

- A **nomenclature section** has been added to the manuscript, listing all relevant variables along with their definitions and units, to facilitate readability.

- The manuscript has been revised to **consistently use full variable names** throughout, particularly in technically dense passages, to improve clarity.

- The **main reference frame** used throughout the study is now explicitly defined in Section 4 as the East-North-Up (ENU) frame, in which all vector quantities (e.g., position, velocity, wind velocity) are expressed. The only exception is the definition of the Euler angles, which follows the North-East-Down (NED) convention to avoid discontinuities in angle representation. Furthermore, Figure 4 has been updated to reflect this clarification, with axis labels revised for improved consistency and readability.

- We have carefully revised the text following the reviewer's 121 in-line comments, aiming to improve clarity and accessibility throughout the manuscript. The response to these comments can be found in the appended pdf.

**Minor Comment 1:**

*"Line 80, The authors claim that their model improves on the existing one by considering the sag of the lines and the dynamics of the KCU. We expect them to present comparisons of measurement results with and without taking these quantities into account. This would make it possible to visualize concretely the impact of taking these factors into account."*

**Response:**

We thank the reviewer for this valuable suggestion. In response, we have revised the text to clarify that the contribution of this work lies in the introduction of a more detailed and physically representative model—rather than claiming direct superiority in estimation performance over previous, simpler models. The proposed model incorporates tether sag and the dynamics of the kite control unit (KCU), which are not commonly accounted for in existing approaches.

While a direct side-by-side quantitative comparison with models omitting these features is beyond the scope of the present study, the advantages of the enhanced model fidelity are illustrated through several key results. In particular, the incorporation of KCU dynamics enables partial estimation of the wing orientation and structural deformation, providing valuable insight into aeroelastic behaviour during flight. These estimations are obtained without substantial increases in computational cost.

To support this point, we have included a new appendix (Appendix C) presenting relevant filter performance metrics and convergence behaviour, which demonstrate the robustness of the implementation.

**Minor Comment 2:**

*"Line 112-114: the KCU appears to be a reference for the orientation of the kite, which does not"*

**Response:**

We appreciate the reviewer's observation. The original formulation may have implied that the KCU serves as a stable reference frame for the orientation of the kite, which is not the case. We have revised the text to clarify that pitch and roll angles are defined with respect to the orientation of the last tether segment, rather than to the KCU. The KCU frame is not used as a reference, as it is not rigidly fixed or well-defined due to its suspended configuration and motion. This correction ensures consistency with the later sections of the manuscript, where kite orientation is explicitly referenced relative to the tether.

**Minor Comment 3:**

*"Line 113-114 The depower angle deserves a more rigorous definition."*

**Response:**

We thank the reviewer for this comment. The text has been revised to provide a clearer and more rigorous definition of the depower angle. We now reference Figure 1 upon first mention and clarify that the depower angle is determined experimentally for each system. Additionally, we have introduced the definition of the tether angle of attack and the parameter $\lambda_0$ to better explain the relationship between the relevant aerodynamic angles.

**Minor Comment 4:**

*"Line 119 Fig.3.a does not exist. Replace by Fig.3"*

**Response:**

We thank the reviewer for noticing the typo, which is now corrected.

**Minor Comment 5:**

*"Line 160 Figure 3 shows 4 boxes on 4 battens. We need to specify which one is the IMU and which other sensors are in the others."*

**Response:**

We thank the reviewer for this observation. We have clarified in the caption that all four battens shown in the image are equipped with GPS+IMU units. Additionally, the specific sensor configurations used in the analysed flights—namely, the number and placement of the units—are now detailed in the results section. Please note that the kite depicted in the figure differs from the one used in the presented datasets.

[revised manuscript text omitted]

---

## Author Response (AR2)

**Response to Reviewer 2, Second Round**

July 8, 2025

1. **It is difficult to interpret the significance of the parameters in Table B2. Could this table be accompanied by a dimensioned drawing to make things clearer for the reader?**

   Thank you for the suggestion. We believe that the existing drawings and sketches already provide sufficient context to interpret the parameters in Table B2. The KCU is modeled as a cylinder, and the distance between the KCU and kite is understood within the modeling framework as the distance between the point masses, since each component is modeled accordingly.

2. **The KCU in Figure 1 does not look like a cylinder of revolution. It therefore seems difficult to reduce its geometry to a 'diameter' and 'length' as shown in Table 2.**

   Thank you for the observation. The KCU in Figure 1 does not include the padding. Otherwise, it resembles more closely a cylinder. Please note that Figure 1 is only a representative illustration. The actual system is shown in Figure 2, where the KCU has a cylindrical cover. Nevertheless, we acknowledge the simplification and note that developing a model for a kite-specific KCU lies outside the scope of this paper.

3. **Furthermore, Figure 1 appears to represent the front and side views of the V3 wing with the bridle system and the KCU. However, the views are not checked for correspondence. The authors need to correct this figure to make it usable for the purposes of reproducibility of the results.**

   Thank you for pointing this out. Figure 1 is a conceptual illustration intended to visualize the setup and components of the kite system. It has been adapted from earlier publications (Oehler and Schmehl, 2019) and is not meant to serve as a technical drawing. For reproducibility purposes, we have added a citation in the figure caption referring to the open-source repository that contains all relevant geometry and measurement data of the V3 kite system.

4. **In the answer to the "Major Concern...", it is mentioned that the kite position and velocity come from the GPS. However, as there are 4 GPS and associated inertial units, the reader can wonder if the result comes from one single GPS or is it a weighted average of the 4 GPSs positions?**

   Thank you for this important clarification request. The number and placement of sensors for each dataset are already described in the results section. For the V3 system, we use the average of the two GPS units installed in the central strut. For the V9 system, only one GPS was typically installed; however, in one of the flights, two GPS units were placed on the central struts, and their positions were likewise averaged. We will clarify this explicitly in the results section.

5. **Are the four GPS relative positions defined based on measurements done on the ground directly on the real kite? Or does the author just rely on the theoretical design and dimension of the kite?**

   Thank you for the question. The datasets studied do not involve four GPS units. Each GPS measures absolute position relative to the ground station. This is already clarified in the results section.

6. **Are the variations in distance between the four boxes due to errors or to kite structure deformations?**

   Thank you for your observation. This is not applicable to the configurations used. The GPS units are not used to measure relative distances; they serve mainly for redundancy.

7. **L200–201: It seems that the definition of lift force forgot the contribution of inertia to the kite motion. Does that mean that the authors assumed it for negligible? This needs to be demonstrated since a priori the authors have the material to do so.**

   Thank you for the remark. Equation 4 defines the aerodynamic force. Inertial effects are included in the equations of motion (see Eq. 23), where the sum of all forces includes inertia. The aerodynamic force itself does not include inertial contributions.

8. **Figure 10(b): The curve for the IMU strut is practically invisible, especially in the bottom figure. Please use different markers and finer lines.**

   Thank you for pointing this out. We have added a note in the figure caption clarifying that the two IMU lines are nearly overlapping and may be difficult to distinguish.

9. **In the same figure, can you replace "aligned with $v_a$" by "$v_a$" for clarity? The same for $v_k$.**

   Thank you for the suggestion. We have updated the legend to use the notation $x_{\mathrm{k}} \parallel v_{\mathrm{a}}$ and $x_{\mathrm{k}} \parallel v_{\mathrm{k}}$, indicating alignment of the kite's body-fixed x-axis (heading) with either the apparent wind velocity or the kite's velocity, respectively. This provides a more concise and consistent representation in the plot.

10. **Lines 500–504: There is confusion with the use of the word "deformation". A modification in wing angle of attack does not necessarily result in deformation. It essentially results in a pitch solid rotation which is quite different.**

    Thank you for the insightful comment. We define the kite as the combination of wing and bridle system. A change in the local angle of attack due to actuation is interpreted as a deformation, since it does not involve rigid body pitch of the entire kite. The solid rotation refers to the bridle–wing assembly around the KCU. From the EKF, it is possible to isolate solid pitch rotation from local deformation, and this is used to estimate the angle of attack at the wing, defined at the central struts of the wing. A small clarification is added to the text.

11. **Lines 522–527: The definition of sag is still unclear to me. For clarity, the author should use Figure 4 to support visually what is "sag". In my mind "sag" is the maximum distance between the deformed tether and the straight line between the tether extremities. Given this definition, talking about negative sag makes no sense.**

    Thank you for pointing this out. In our work, what we initially referred to as "sag" is defined as the difference between the unstretched tether length and the radial distance (i.e., the straight-line distance between tether attachment points). Since this quantity can be negative in tensioned configurations, we acknowledge that the term "sag" may be misleading. We will revise the text to use the term "tether slack" instead and clarify the definition accordingly in the main text and figure.

12. **Line 549 (and others 591, 593. . . ): It might be useful to cite references other than those from Delft.**

    Thank you for the suggestion. In this section, we specifically discuss kite configurations developed and studied at TU Delft. There are currently no other institutions with equivalent datasets or results for these specific systems.

13. **Lines 551–554: The text is unclear. It is necessary to specify which figure is being discussed. The comments should be more precise and clearer.**

    Thank you for pointing this out. We have revised the text to explicitly refer to Figure 14 and made the explanation more precise for improved clarity.

14. **Line 784: "A standard laptop" – can you give the main characteristics (RAM, cores, CPU, etc.)? A reference to Annex C2 may be sufficient.**

    Thank you. The hardware and software specifications of the standard laptop used for benchmarking are already provided in Appendix C2, and the reference was included in the text. To make this clearer, we have slightly adjusted the sentence to explicitly refer to Appendix C2 directly after mentioning the standard laptop.

---

## Author Response (AR3)

**Response to Associate Editor Comments**

July 18, 2025

We thank the associate editor for the constructive and detailed feedback. Below, we address each point in turn:

1. **Terminology.** *It is more appropriate to use "vertical velocity" or "vertical velocity component" rather than "vertical wind speed."*
   We revised the terminology throughout the manuscript accordingly.

2. **PSD in Fig. 23.** *The PSD appears excessively noisy. Consider using Welch's method with overlapping segments.*
   We re-computed the PSD using Welch's method with three segments and 50% overlap. The result is now significantly less noisy and the $-5/3$ slope region is more visible.

3. **Figure 23 – Units.** *The vertical axis is missing the unit "Hz". Use "$m^2/s^2$ per Hz" or "$m^2$ $s^{-2}$ $Hz^{-1}$."*
   We corrected the label to "$m^2$ $s^{-2}$ $Hz^{-1}$".

4. **Figure font size and readability.** *Check that all fonts are legible in two-column format.*
   We revised all figures to ensure consistent and legible font sizes in the final two-column layout.

5. **Figures 8 and 24 – Panel references.** *Move sub-panel references (e.g., (a), (b)) into the figure captions.*
   We updated the captions to include the sub-panel references.

6. **Figure captions – Completeness.** *Some captions are too brief (e.g., Fig. 7). Consider including date and time.*
   We expanded several captions to improve clarity and added relevant contextual information to Fig. 7.

7. **Figure 13 – Axis label.** *Consider adding "$C_d$" or "Drag coefficient" to the y-axis label.*
   We added "$C_d$" to the y-axis label of the middle panel in Fig. 13.

8. **Units – Formatting.** *Ensure correct spacing in "$m$ $s^{-1}$" to avoid confusion with milliseconds.*
   We corrected all unit formatting to include the space: "$m$ $s^{-1}$".

9. **Table C1 – Unit column.** *Add a column for units, consistent with Tables B2–B5.*
   We added a "Unit" column to Table C1 for consistency.

10. **References – Fagiano et al.** *Consider citing only the 2014 paper.*
    We removed the 2013 conference paper and retained only the 2014 journal publication.

11. **Figure 3.** *Optional: Indicate the GPS+IMU units, Pitot tube, and wind vanes with arrows.*
    We added visual indicators (circles) in the image, and clarified them in the caption.

We trust these changes address all points satisfactorily and further improve the manuscript's clarity and consistency. We appreciate the associate editor's valuable feedback.